# Why Transformers Need Adam: A Hessian Perspective

**Yushun Zhang**[13]**, Congliang Chen**[13]**, Tian Ding**[23]**, Ziniu Li**[13]**, Ruoyu Sun**[123*]**, Zhi-Quan Luo**[13]

[1]The Chinese University of Hong Kong, Shenzhen, China
[2]Shenzhen International Center For Industrial And Applied Mathematics, Shenzhen, China
[3]Shenzhen Research Institute of Big Data, Shenzhen, China
`{yushunzhang,congliangchen,ziniuli}@link.cuhk.edu.cn`
`dingtian@sribd.cn, sunruoyu@cuhk.edu.cn, luozq@cuhk.edu.cn`

## Abstract

SGD performs worse than Adam by a significant margin on Transformers, but the reason remains unclear. In this work, we provide an explanation through the lens of Hessian: (i) Transformers are "heterogeneous": the Hessian spectrum across parameter blocks vary dramatically, a phenomenon we call "block heterogeneity"; (ii) Heterogeneity hampers SGD: SGD performs worse than Adam on problems with block heterogeneity. To validate (i) and (ii), we check various Transformers, CNNs, MLPs, and quadratic problems, and find that SGD can perform on par with Adam on problems without block heterogeneity, but performs worse than Adam when the heterogeneity exists. Our initial theoretical analysis indicates that SGD performs worse because it applies one single learning rate to all blocks, which cannot handle the heterogeneity among blocks. This limitation could be ameliorated if we use coordinate-wise learning rates, as designed in Adam.[1]

## 1 Introduction

Transformers [83] have become a major workhorse behind AI development (e.g., [1]). However, the understanding of Transformer training remains limited. For instance, Transformer training largely relies on the Adam optimizer [45, 57]. In contrast, stochastic gradient descent with momentum (SGD)[2], the de-facto optimizer for convolution neural networks (CNNs) [49], performs significantly worse than Adam on Transformers (e.g., Figure 3). Yet, the reasons behind this performance gap remain unclear. Understanding why SGD performs worse than Adam on Transformers is an intriguing question. **First,** from a theoretical perspective, this can help us better understand the training of Transformers and more generally, neural networks. **Second,** from a computational perspective, the understanding may inspire the design of better algorithms for training neural networks.

In this work, we explore why SGD largely underperforms Adam on Transformers through the lens of Hessian. We start by investigating the *full* Hessian spectrum of Transformers, i.e., the full eigenvalue density of Hessian (see Figure 1). By theory, the full Hessian spectrum largely determines the behavior of gradient-based methods [63, 33, 79, 37], so we suspect it may also help explain SGD's unsatisfactory performance. Using tools from numerical linear algebra [7], we empirically compare the full spectra of CNNs (where SGD is on par with Adam) and those of Transformers (where SGD largely lags behind Adam). Unfortunately, as shown in Figure 1, the spectra for CNNs and Transformers are often largely *similar* despite the different optimizer behaviors. As such, we have *not* identified critical features in the full Hessian spectra associated with the gap between Adam and SGD on Transformers. To reveal the cause, a more fine-grained investigation into the Hessian is needed.

---

[*]: Correspondence author.
[1]Our code is available at `https://github.com/zyushun/hessian-spectrum`.
[2]We introduce the update rules of Adam(W) and SGD in Appendix C.1.

38th Conference on Neural Information Processing Systems (NeurIPS 2024).

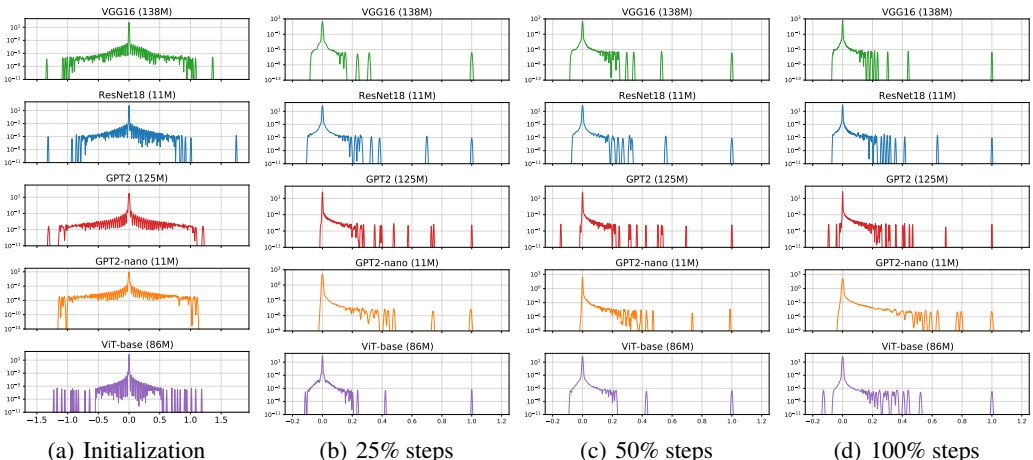

Figure 1: The full Hessian spectra of CNNs (VGG16 and ResNet18) and Transformers (GPT2, GPT2-nano, and ViT-base) at different training stages. The $x$-axis records the eigenvalues and the $y$-axis records the frequency in the log scale. To allow comparison in the same figure, the plotted spectra are normalized by their 10th largest eigenvalues. We find that the spectra on CNNs and Transformers are largely similar.

What would cause SGD to perform significantly worse than Adam on Transformers, but not on CNNs? By dissecting the structures of CNNs and Transformers, we notice that CNNs are constructed by the repetitive stacking of *similar* parameter blocks (convolution layers), while Transformers involve the non-sequential stacking of *disparate* parameter blocks (e.g. Query, Key, Value, Output projection blocks in attention and MLP layers). We hypothesize that these architectural differences might lead to different optimization properties. Intuitively, disparate parameter blocks contribute differently to the overall loss. So each block might benefit from a specialized treatment by optimizers, a flexibility offered by Adam but not by SGD. This observation motivates us to investigate the Hessian spectrum of each parameter block, which we refer to as the blockwise Hessian spectrum.

By inspecting the blockwise Hessian spectrum, we discover a possible explanation for why SGD is worse: the "heterogeneity" inherent in Transformers. We provide both empirical and theoretical evidence to support this explanation. Our contributions can be summarized as follows:

- **Why SGD underperforms Adam on Transformers.** We explain why SGD is worse than Adam on Transformers by examining the blockwise Hessian spectrum. First, we identify a phenomenon called "block heterogeneity", which refers to the large differences in the Hessian spectra across parameter blocks. This block heterogeneity is observed in all examined Transformers but not in CNNs. Second, we verify that block heterogeneity hinders SGD. Across various Transformers, CNNs, and MLPs, we show that SGD consistently performs worse than Adam on problems with block heterogeneity but can perform similarly to Adam, otherwise.

- **Theoretical results on quadratic models.** We construct convex quadratic problems with and without block heterogeneity and find that gradient descent (GD) largely underperforms Adam on problems with block heterogeneity, but can perform comparably otherwise. Our theoretical analysis shows that GD can be slower than Adam on quadratic problems with block heterogeneity. We point out GD is slower than Adam because it uses a single learning rate for all blocks. The deficiency can be mitigated by assigning different learning rates across blocks, as Adam does.

We emphasize that we do *not* claim block heterogeneity is the only cause for the performance gap between Adam and SGD, but just that it is at least one important cause. We verify, both empirically and theoretically, that SGD underperforms Adam when block heterogeneity is present.

## 2 Problem Settings and Initial Attempts

### 2.1 Problem Settings

**Notations.** We denote the training loss as $\mathcal{L}(w)$, where $w \in \mathbb{R}^d$ is the neural network parameters. We denote the gradient and Hessian of the training loss w.r.t. neural network parameters as $\nabla \mathcal{L}(w) \in \mathbb{R}^d$ and $\nabla^2 \mathcal{L}(w) \in \mathbb{R}^{d \times d}$, respectively. We use $[d]$ to denote the index set $\{1, 2, \cdots, d\}$. Given an arbitrary partition $\{\mathcal{D}_l\}_{l=1}^L$ over $[d]$ with $d_l \triangleq |\mathcal{D}_l|$, we can split $w$ into $L$ parameter blocks $\{w_l\}_{l=1}^L$, where $w_l = \mathbb{R}^{d_l}$ consists of parameters with indexes in the $l$-th block $\mathcal{D}_l$. We denote $[\nabla^2 \mathcal{L}(w)]_l \in$

$\mathbb{R}^{d_l \times d_l}$ as the Hessian of $l$-th parameter-block $w_l$, where $[\nabla^2 \mathcal{L}(w)]_{l,i,j} = \frac{\partial^2}{\partial_{w_{l,i}} \partial_{w_{l,j}}} \mathcal{L}(w_l)$. Note that $[\nabla^2 \mathcal{L}(w)]_l$ is the $l$-th principal block sub-matrix of $\nabla^2 \mathcal{L}(w)$.

**Setups.** Hessian of large-scale NNs are intractable to compute and store. In this work, we apply a numerical tool called Stochastic Lanczos Quadrature method (SLQ) [7] to approximate the Hessian spectrum. SLQ uses a smooth curve on $\mathbb{R}$ to approximate the histograms of eigenvalues (see Figure 1 as an example). A detailed introduction to SLQ is provided in Appendix C.2. All experimental setups are shown in Appendix D. We focus primarily on the following models/tasks.

- **CNNs.** We study ResNet18 (11M) and VGG16 (138M) on ImageNet [40, 78]. On these tasks, SGD performs on par with Adam. **See Figure 9 in Appendix B for the evidence.**

- **Transformers.** We study Transformer with various scales and modalities, including GPT2 (125M) on OpenWebText [71]; ViT-base (86M) on ImageNet [27]; BERT (40M) on Cornell Movie-Dialogs Corpus [25]; GPT2-nano[3] (11M) on English corpus. On these tasks, SGD performs significantly worse than Adam. **See Figure 10 in Appendix B for the evidence.**

For each model, we estimated **(1)** the full Hessian spectrum $\nabla^2 \mathcal{L}(w)$, and **(2)** the blockwise Hessian spectrum $[\nabla^2 \mathcal{L}(w)]_l, l \in [L]$. For the latter, we split $w$ according to the default partition in PyTorch implementation, e.g., Embedding layer, Query in each attention layer, Key in each attention layer, Value in each attention layer, etc. Note that the term "block" differs from the term "layer". For instance, Query and Key can reside in the same layer but are different parameter blocks.

## 2.2 Full Hessian Spectrum Is Not Informative Enough

We study the full Hessian spectrum of Transformers for two reasons. First, as stated in Section 1, the Hessian spectrum significantly influences the behavior of gradient methods [63]. Second, previous research shows that the Hessian spectrum provides insights into neural network phenomena, like BatchNorm's effect on training speed [32]. Therefore, we hypothesize that the Hessian spectrum may also explain why SGD largely lags behind Adam on Transformers.

We compare the full Hessian spectra of CNNs (where SGD performs similarly to Adam) and those of Transformers (where SGD underperforms Adam), as shown in Figure 1. Unfortunately, the results suggest that the full Hessian spectrum alone may not suffice to explain the gap between Adam and SGD on Transformers. We elaborate as follows. The primary information in the spectrum lies in its (A) dispersion, (B) shape, and (C) evolution during training. Regarding (A), we observe that the eigenvalues are dispersed similarly across different models, with no notably large outlier for Transformers. Thus, dispersion does not seem to be related to why SGD is worse than Adam. We further investigate (B) and (C). For all CNNs and Transformers in Figure 1, we observe similar phenomena: the spectrum's shape is approximately symmetrical around 0 at initialization. As training proceeds, the majority of negative eigenvalues disappear, and the shape evolves into a combination of a "bulk" and some "outliers". Since the spectral shape and evolution are quite similar for both Transformers and CNNs, they cannot explain why SGD is worse than Adam on Transformers, either. In summary, we have not identified any critical phenomena in the full Hessian spectra that can be linked to the performance gap between Adam and SGD on Transformers.

## 2.3 Motivations of Investigating Blockwise Hessian Spectra

What other factors could cause SGD to perform significantly worse than Adam on Transformers but not on CNNs? We identify one critical feature that has been overlooked in the full Hessian spectrum analysis above: **the building-up rules of Transformers.** As shown in Figure 3, CNNs are constructed by the repetitive stacking of *similar* parameter blocks (convolution layers). In contrast, Transformers consist of *disparate* parameter blocks, e.g. Query, Key, Value in attention, and MLP layers. Further, these blocks are stacked in a non-sequential manner. We hypothesize that the "different designs among parameter blocks" can be reflected in the Hessian of these parameter blocks, which might affect algorithmic behavior. This inspires us to investigate **the blockwise Hessian spectra**, i.e., the spectrum of principal blocks of Hessian $[\nabla^2 \mathcal{L}(w)]_l, l \in [L]$.

In parallel to the motivation above, we further provide another evidence that blockwise spectra might be helpful. Classical literature showed that the Hessians of neural nets are *near-block-diagonal matrices* [18], i.e., the magnitudes in the Hessian principle blocks are much larger than those in the off-diagonal blocks. We restate their findings in Figure 2 (a). This implies that the majority of

---

[3] https://github.com/karpathy/nanoGPT/

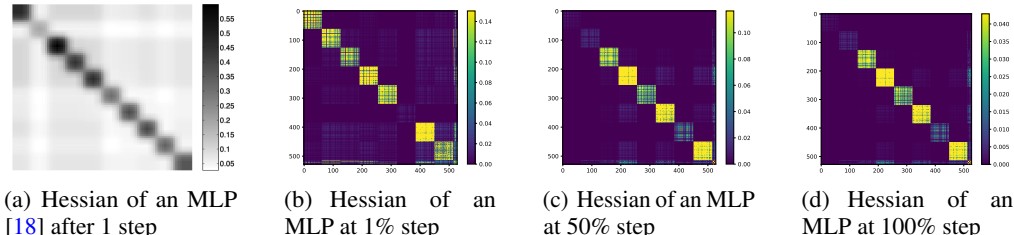

(a) Hessian of an MLP [18] after 1 step

(b) Hessian of an MLP at 1% step

(c) Hessian of an MLP at 50% step

(d) Hessian of an MLP at 100% step

Figure 2: **(a)**: The Hessian of an MLP after 1 training step reported in [18]. **(b,c,d)**: We calculate the Hessians of an MLP (with 8 neurons) at different training stages. We find the near-block-diagonal structure maintains along the training.

Hessian information indeed lies in its principle blocks, and the blockwise Hessian of neural nets might contain valuable information.

To summarize, the "heterogeneous" building-up rules of Transformers inspire us to check the block-wise Hessian, i.e., the principle blocks of the Hessian. The classical results of neural nets [18] further support us to explore this direction since they find that the majority of Hessian information indeed lies in its principle blocks. In the following, we study the blockwise Hessian spectra of various neural networks. For ease of implementation, we define parameter blocks under the PyTorch partition. We show that the blockwise spectra indeed carry more information than the full spectrum for distinguishing CNNs and Transformers.

**Remark: why near-block-diagonal?** We briefly restate the analysis in [18, Section 7] to explain the near-block-diagonal Hessian structure of neural nets [4]. Consider minimizing $\ell(f(\theta, x), y)$ where $\ell(\cdot, \cdot)$ is the Cross-Entropy (CE) loss, $f(\theta, x) = \sum_{i=1}^{n} v_i \phi(w_i^\top x)$ is an 1-hidden-layer neural network with input $x \in \mathbb{R}^d$, weight $w_i \in \mathbb{R}^d$, $v_i \in \mathbb{R}$, and label $y \in \{0, 1\}$, then the off-diagonal-block Hessian elements will contain

$$\frac{\partial^2 \ell(f(\theta, x), y)}{\partial w_i \partial w_j} = p_\theta(y|x)\left(1 - p_\theta(y|x)\right) v_i v_j \phi'\left(w_i^\top x\right) \phi'\left(w_j^\top x\right) x x^\top \quad \text{for } i \neq j, \tag{1}$$

where $p_\theta(y|x) = 1/(1 + \exp(-yf(\theta, x)))$ and $\phi'(\cdot)$ is the derivative of $\phi(\cdot)$. Note that the term $p_\theta(y|x)\left(1 - p_\theta(y|x)\right)$ will vanish rapidly since the training objective is to maximize $p_\theta(y|x)$. Consequently, this drives the Hessian towards a near-block-diagonal configuration, with each block representing an output neuron. This result is validated in Figure 2: we find that the near-block-diagonal structure appears at 1% step and it maintains along the training.

## 3 Main Results

### 3.1 Transformers Exhibit Block Heterogeneity in Hessian, while CNNs Do Not

We now compare the shape of blockwise spectra in VGG16 [40] (CNN) and BERT [25] (Transformer). We sample four blocks for each model and present the spectra in Figure 3. In BERT, the spectra of embedding, attention, and MLP blocks are largely *different*. In contrast, in ResNet, the spectra of convolution layers are *similar*. We further verify this observation for the rest of the parameter blocks. We calculate the Jensen-Shannon (JS) distance between two eigenvalue densities of all possible block pairs and show the results in Figure 4. We summarize our findings in **Observation 1**.

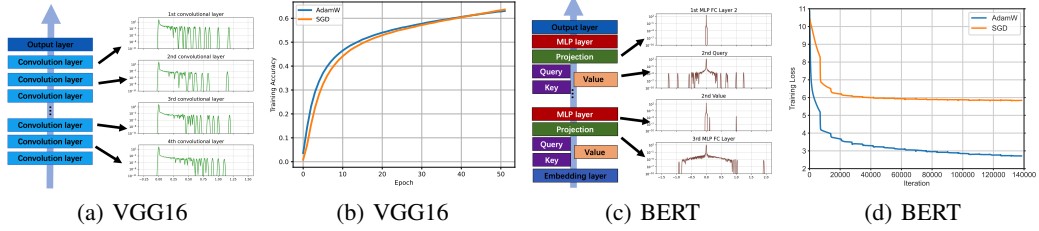

(a) VGG16

(b) VGG16

(c) BERT

(d) BERT

Figure 3: **(a) (c)**: The blockwise Hessian spectra of VGG16 (CNN) and BERT (Transformer) at initialization. The $x$-axis records the eigenvalues and the $y$-axis records the frequency in the log scale. To allow comparison in the same figure, we sample 4 blocks in each model. The plotted spectra are normalized by their 10th largest eigenvalues. The spectra are similar among blocks for VGG and differ significantly across blocks for BERT. **(b) (d)** Adam v.s. SGD for training VGG16 and BERT.

---

[4]We note that this analysis is rather informal; a rigorous theoretical study is left for future work.

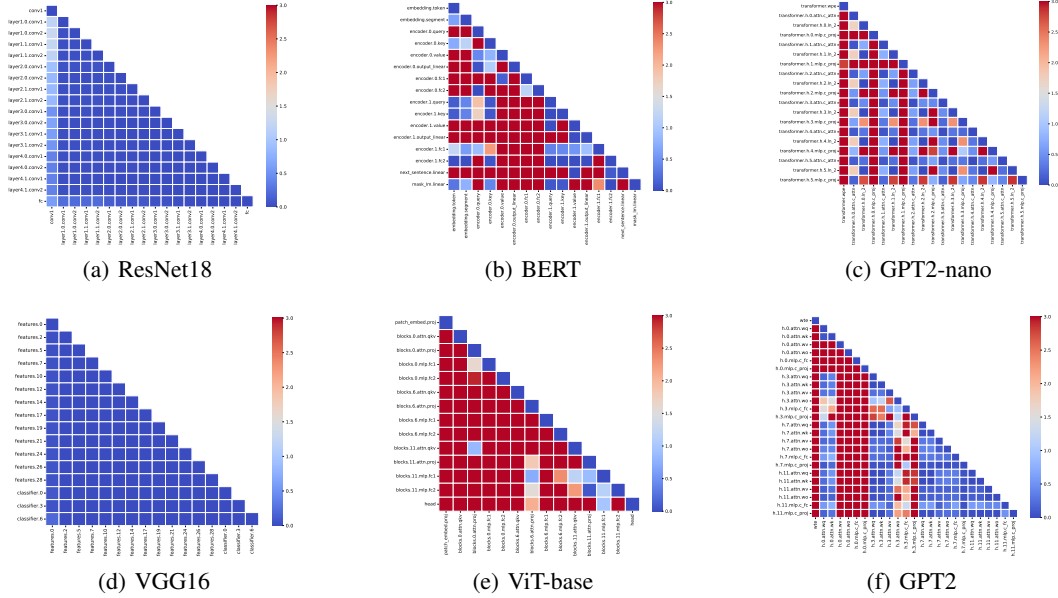

(a) ResNet18      (b) BERT      (c) GPT2-nano

(d) VGG16      (e) ViT-base      (f) GPT2

Figure 4: The JS distance among blockwise Hessian spectra at initialization. We find that the JS distance of blockwise spectra in CNNs is significantly smaller than that in Transformers.

**Observation 1:** For all Transformers we checked, the blockwise Hessian spectra are largely *different* from each other. In contrast, the blockwise Hessian spectra of CNNs are *similar*.

In the following, we refer to the phenomenon of Transformers as "**block heterogeneity**", and refer to that of CNN as "**block homogeneity**". The observations in Figure 3 and 4 indicate that block heterogeneity is informative in distinguishing CNNs and Transformers. In the following, we will show that the block heterogeneity is strongly correlated with the performance gap between SGD and Adam on Transformers.

### 3.2 SGD Performs Worse than Adam on Various Tasks with Block Heterogeneity

Figure 3 and 4 have shown that (**1**) SGD is worse than Adam on Transformers. (**2**) Transformers have block heterogeneity. Now we further link block heterogeneity to SGD's unsatisfactory performance on **non-Transformer** models. This would directly establish a connection between "block heterogeneity" and "why SGD is worse than Adam", without going through Transformers or attention blocks as an intermediary. We consider one man-made example and one real-world example.

**Example 1: A man-made MLP.** We consider a 4-layer MLP on MNIST and change the degree of heterogeneity by scaling each layer by constant $c$. Figure 5 (a) shows SGD gradually performs worse than Adam as heterogeneity grows.

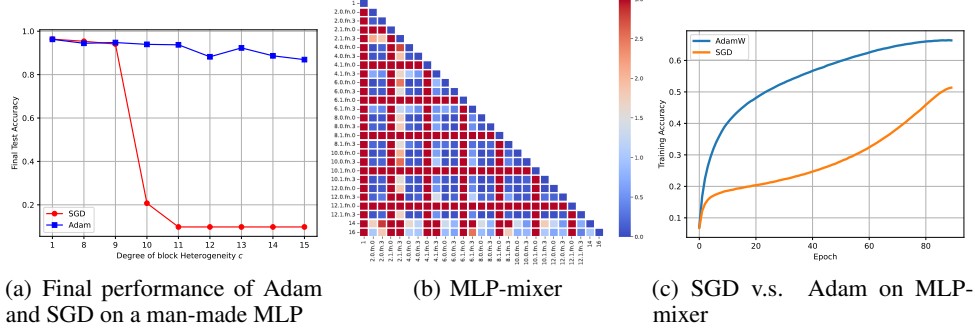

(a) Final performance of Adam and SGD on a man-made MLP      (b) MLP-mixer      (c) SGD v.s. Adam on MLP-mixer

Figure 5: (a) SGD v.s. Adam on a man-made MLP with different degrees of heterogeneity $c$. Each point records the best-converged test accuracy under the learning rate grid search. SGD performs worse as heterogeneity grows. (b) The JS distance among blockwise Hessian spectra for MLP-mixer [81] at initialization. We observe heterogeneity. (c) SGD performs worse than Adam on MLP-mixer.

**Example 2: MLP-mixer.** We consider MLP-mixer [81], a famous all-MLP architecture that outperforms CNNs and ViTs on some vision tasks. Figure 5 (b) (c) show that the initial Hessian of MLP-mixer has block heterogeneity and SGD lags behind Adam on this architecture.

We summarize the findings so far in **Observation 2**.

> **Observation 2:** For all tasks that we checked, SGD is worse than Adam when block heterogeneity exists, regardless of whether Transformers or attention mechanisms are utilized.

### 3.3  Reduced Block Heterogeneity in Pre-trained Transformers

We remark that different Transformers exhibit different levels of block heterogeneity. Although all examined Transformers show strong block heterogeneity, we find that this heterogeneity can be mitigated, resulting in less performance deterioration for SGD. As illustrated in Figure 6, pre-trained GPT2 on SFT tasks can exhibit less block heterogeneity compared to pre-training GPT2 from scratch (Figure 4 (f)). In this case, although SGD is still slower than Adam, it achieves a similar loss at convergence. Compared with training GPT2 from scratch (Figure 10 (d) in Appendix B), the performance gap between SGD and Adam is significantly narrowed down. These findings suggest that the heterogeneity induced by architectural design can be alleviated by selecting "good" weights. This partly explains why simpler methods like SGD and even its zeroth-order version can still be effective for fine-tuning language models, albeit with slower convergence [59, 60].

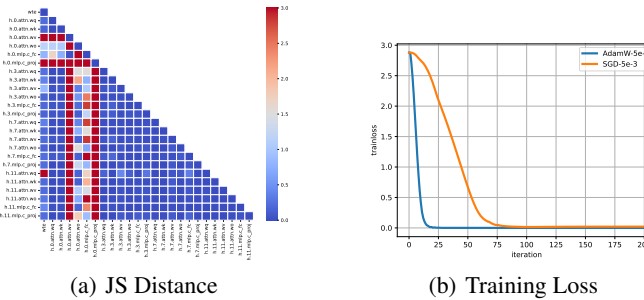

(a) JS Distance  (b) Training Loss

Figure 6: We fine-tune GPT2 (pre-trained) on Alpaca Eval, and plot (a) the JS distance among blockwise Hessian spectra; (b) the training loss of SGD and Adam.

In Figure 13 in Appendix B, we further report the evolution of the block heterogeneity of ViT-base along the training. Similarly to GPT2 in Figure 6, we find that the block heterogeneity of ViT-base tends to reduce after the training. In addition, we find that SGD can perform better when initializing at the weight with less heterogeneity, e.g., initializing at 50% total training steps. We hypothesize that "the attenuation of Hessian heterogeneity" is a common phenomenon after training, and we leave detailed investigation as a future direction.

> **Observation 3:** Block heterogeneity in Hessian tends to reduce after (pre)-training.

### 3.4  Implication on Choosing SGD or Adam

We have shown that SGD can largely underperform Adam on various architectures. This leads to an intriguing question: **Can we predict the incompetence of SGD before the training begins**?

Our findings can bring up an empirical guidance: we can compute the blockwise spectrum of initial Hessian, and then decide whether to use Adam or SGD. Such a method could be useful in scenarios in training large models that are not mainstream Transformers or CNNs, e.g., Mamba [38]. In these cases, there is not much prior experience in choosing optimizers. It would be intriguing to decide whether SGD is suitable for the task before the training is launched. One might argue that simple trial is enough: try both SGD and Adam; if Adam is remarkably better, then pick Adam; if Adam and SGD are similar, then pick SGD. Nevertheless, this simple approach may not be easy for large models. First, for large models, it may take days to know one run of an algorthm is good or not. Second, it requires tuning hyperparameters at least a few times to get a reasonably good judgement, making the cost of trial even higher.

We here propose a quantitative metric that could predict the incompetence of SGD before the training. With the help of this metric, we could save much expense on the trial and error for SGD. The metric is simply the averaged JS distance among blockwise Hessian spectra at initialization, i.e., the averaged value in the heatmap of Figure 4. We denote it as $JS^0$. We present $JS^0$ of various models in Table 1. Note that $JS^0$ establishes a quantitative difference between the loss landscape of Transformers and CNNs. Further, $JS^0$ is independent of optimizers and could be checked before training.

Table 1: $JS^0$ denotes the average JS distance between the initial Hessian spectra of each pair of parameter blocks. A larger $JS^0$ suggests that the task is more difficult for SGD.

| Model | ResNet18 | VGG16 | GPT2 (pretrained) | MLP-mixer | BERT | GPT2 | ViT-base |
|-------|----------|-------|-------------------|-----------|------|------|----------|
| $JS^0$ | 0.10 | 0.09 | 18.84 | 34.90 | 53.38 | 83.23 | 286.41 |

To validate the effectiveness of the quantitative metric $JS^0$, we summarize $JS^0$ of different models and the corresponding SGD performance in Figure 7. We find that the performance gap between SGD and Adam becomes greater as $JS^0$ increases. Thus, $JS^0$ can serve as a potential indicator to predict whether SGD may underperform Adam.

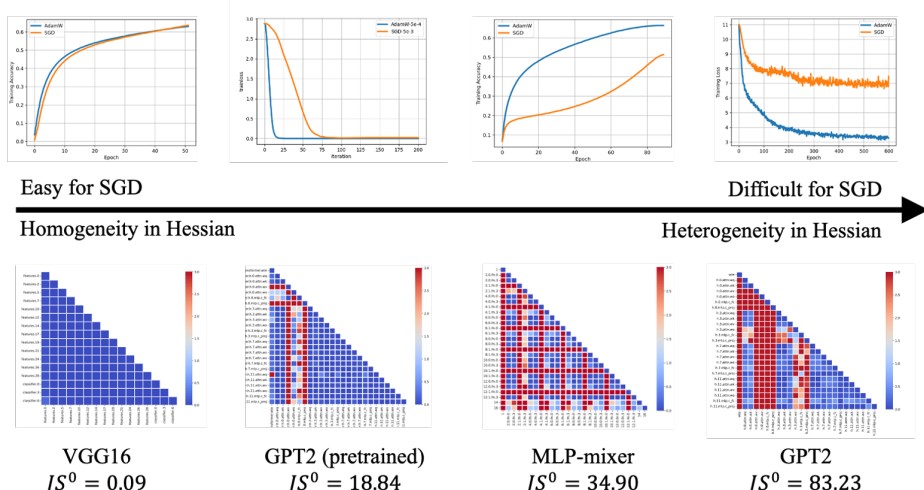

Figure 7: Comparison of $JS^0$ and the performance of SGD on different models. We find the performance gap between SGD and Adam becomes greater as $JS^0$ increases.

Finally, we remark $JS^0$ is rather expensive to compute due to the overhead of SLQ: it requires comparable time to one training run. Fortunately, we find the original SLQ is rather redundant for measuring hessian heterogeneity. We propose some simple tricks to significantly reduce the computation time, while still effectively detecting the Hessian heterogeneity. We call it simplified SLQ and we present it in Table 3 in Appendix B. As a result, the simplified SLQ can obtain the same message as in Table 1 while only taking negligible time (e.g., $< 0.001$s for ResNet18).

## 4   Case Study of Quadratic Models and Preliminary Theory

Now we study quadratic functions with block diagonal Hessian, with or without block heterogeneity. Note that insights on quadratic models could be important for understanding realistic NNs, as mentioned by researchers such as LeCun et al. [50] and OpenAI team [44].

**Setups and additional notations.** We consider the following quadratic minimization.

$$\min_{w \in \mathbb{R}^d} \mathcal{L}(w) = \frac{1}{2} w^T H w - h^T w,$$

where $H \in \mathbb{R}^{d \times d}$ is positive definite and $h \in \mathbb{R}^d$. We denote $\mathcal{L}^*$ as the minimum value of $\mathcal{L}(w)$. We set $H$ as a block diagonal matrix: $H = \text{diag}(H_1, \cdots, H_L)$, where $H_l \in \mathbb{R}^{d_l \times d_l}$ and $d = \sum_{l=1}^{L} d_l$. We use $w_l \in \mathbb{R}^{d_l}$ to denote the variable in the $l$-th block and $w = (w_1^T, \cdots, w_L^T)^T \in \mathbb{R}^d$. Similarly for $h_l \in \mathbb{R}^{d_l}$. Similarly, we use $[\nabla L(w)]_l \in \mathbb{R}^{d_l}$ to denote the gradient in the $l$-th block and denote $[\mathcal{L}(w)]_l = \frac{1}{2}(w_l^t)^T H_l w_l^t - h_l^T w_l$ as the objective function w.r.t. the $l$-th block. Note that $\mathcal{L}(w) = \sum_{l=1}^{L} [\mathcal{L}(w)]_l$. We denote $\lambda_1 \geq \lambda_2 \cdots \geq \lambda_d$ as the eigenvalues of $H$. Similarly for $\lambda_{l,1} \cdots \lambda_{l,d_l}$. We denote $\kappa = \frac{\lambda_1}{\lambda_d}$ and $\kappa_l = \frac{\lambda_{l,1}}{\lambda_{l,d_l}}$ as the condition number of $H$ and $H_l$, respectively. We say an algorithm has complexity $\tilde{\mathcal{O}}(C)$ if it takes $\mathcal{O}(C \log(1/\epsilon))$ iterations to achieve error $\frac{\mathcal{L}(w) - \mathcal{L}^*}{\mathcal{L}(w^0) - \mathcal{L}^*} \leq \epsilon$, where $w^0$ is the initial point.

### 4.1   Experimental Observations

We consider four types of Hessian $H$ as follows. For all cases, we set condition number = 5000.

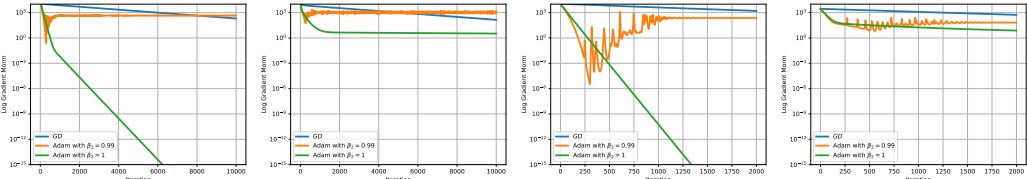

(a) Hessian with GPT2 block-wise spectrum  (b) Hessian with ResNet18 blockwise spectrum  (c) Hessian with simplified heterogeneous blocks  (d) Hessian with simplified homogeneous blocks

Figure 8: The performance of Adam and GD on homo/heterogeneous quadratic problems. The condition numbers of Hessian equal to 5000 for all four cases. When blocks are heterogeneous, GD largely lags behind Adam, and GD performs similarly to Adam if otherwise.

- **Case 1: Hessian with Transformer-type spectra.** We choose $L = 4$ and $d_l = 25$. For $l \in [L]$, we construct $H_l = Q_l \Lambda_l Q_l^T$ where $Q_l$ are matrices with i.i.d. standard Gaussian entries and $\Lambda_l$ are diagonal matrices. For the diagonal elements in $\Lambda_l$, we sample $d_l$ numbers according to the spectrum of the embedding layer; 3rd Query, 3rd Value, 3rd MLP (`fc` layer) in GPT2. Shifting and proportional scaling are performed to ensure all elements in $\Lambda_l$ lie in the interval $[1, 5000]$. This ensures strong convexity and controls the condition number of $H$ equals 5000. The spectra of $H_l$ are in Figure 14 in Appendix B. We choose $h = 0$ for all cases.
- **Case 2: Hessian with CNN-type spectra.** We use the same setup as in **Case 1**. For the diagonal elements in $\Lambda_l$, we sample $d_l$ numbers according to the spectrum of the 1st to 4th convolution layers in ResNet18. We then shift and scale $\Lambda_l$ to the interval $[1, 5000]$ to ensure strong convexity and a condition number of 5000. The spectra of $H_l$ are shown in Figure 15 in Appendix B.
- **Case 3: Hessian with simplified heterogeneous spectra.** We choose $L = 3$ and $d_l = 3$. For $l \in [L]$, we construct $H_l = Q_l \Lambda_l Q_l^T$ where $Q_l$ are independent standard Gaussian random matrix and $\Lambda_l$ are diagonal matrices. We set the diagonal elements of $\Lambda_l$ as $\{1, 2, 3\}, \{99, 100, 101\}, \{4998, 4999, 5000\}$ for $l = 1, 2, 3$, respectively. The spectra of $H_l$ are different due to their different supports. The condition number of Hessian $H$ is 5000.
- **Case 4: Hessian with simplified homogeneous spectra.** We consider the same setup as **Case 3**. We set the diagonal elements of $\Lambda_l$ as $\{1, 99, 4998\}, \{2, 100, 4999\}, \{3, 101, 5000\}$ for $l = 1, 2, 3$, respectively. The spectra of $H_l$ are similar. The condition number is 5000.

Now we study two types of optimizers: one that assigns a single learning rate for all blocks, and one that assign different learning rates across blocks.

- **Single-learning-rate optimizer.** We study gradient descent (GD).

$$w^{t+1} = w^t - \eta \nabla \mathcal{L}(w) = w^t - \eta(Hw^t - h) \tag{2}$$

  We use the optimal learning rate $\eta = \frac{2}{\mu + L}$ [63]. We use standard Gaussian initialization.
- **Coordinate-wise-learning-rate optimizer.** We study Adam with a constant learning rate and with no bias correction for simplicity (Algorithm 3). We set $\beta_1 = 0$ to erase the effect of momentum. **This helps us to focus on the effect of coordinate-wise learning rate** (or the effect of diagonal preconditioning) in Adam. We use $\epsilon = 0$. We consider $\beta_2 = 1$ and $\beta_2 = 0.99$, respectively. When $\beta_2 = 1$, Adam assigns coordinate-wise learning rates according to the initial gradient, but these learning rates are fixed along iteration. The update rule is as follows.

$$w^{t+1} = w^t - \eta(D^0_{Adam})^{-1} \nabla \mathcal{L}(w) = w^t - \eta(D^0_{Adam})^{-1}(Hw^t - h), \tag{3}$$

  where $D^0_{Adam} = \mathrm{diag}(\nabla \mathcal{L}(w^0) \circ \nabla \mathcal{L}(w^0))^{\frac{1}{2}}$ and $\nabla \mathcal{L}(w^0) = Hw^0 - h$. When $\beta_2 < 1$, the coordinate-wise learning rates adaptively change along iteration. The update rule is as follows (note that $\nabla \mathcal{L}(w^k) = Hw^k - h.$).

$$w^{t+1} = w^t - \eta(D^t_{Adam})^{-1} \nabla \mathcal{L}(w) = w^t - \eta(D^t_{Adam})^{-1}(Hw^t - h), \quad \text{where} \tag{4}$$

$$D^t_{Adam} = \mathrm{diag}\left( (1 - \beta_2) \left( \sum_{k=1}^{t} \beta_2^{t-k} \nabla \mathcal{L}(w^k) \circ \nabla \mathcal{L}(w^k) \right) + \beta^t \nabla \mathcal{L}(w^0) \circ \nabla \mathcal{L}(w^0) \right)^{\frac{1}{2}}$$

  We grid search $\eta$ and use the standard Gaussian initialization. We remark that when $\beta_2 < 1$, Adam would bounce among non-optimal points. This will be shown in Proposition 2.

**Summary of experimental observations.** Figure 8 presents two phenomena. For Hessian with heterogeneous blocks (**Case 1 and 3**), GD largely lags behind Adam. For Hessian with homogeneous

blocks (**Case 2 and 4**), GD is on par with Adam. We emphasize that all Hessians have the same condition number. Further, Hessian in **Case 3** and **4** share all the eigenvalues (not just the extreme ones). The gap between Adam and GD is purely due to the different blockwise spectra caused by the different locations of eigenvalues. **Case 3** and **4** help reveal the causal relation between "block heterogeneity in Hessian" and "GD is worse than Adam". We hypothesize that GD performs badly because it uses one single learning rate for all blocks, which cannot handle the heterogeneity among blocks. Such heterogeneity can be better handled using different learning rates across blocks, as designed in Adam.

### 4.2 Initial Theoretical Results

We now provide initial theoretical results to characterize how GD lags behind Adam in problems with heterogenous Hessian. Note that classical optimization theory depicts the rate of first-order methods by the condition number of the full Hessian $\kappa$. However, we point out that $\kappa$ is not informative enough to describe the performance gap in Figure 8 since $\kappa$ is the same in all four cases. To distinguish Adam and GD, we need to utilize more fine-grained quantities like blockwise spectra of sub-matrices.

Note that the blockwise spectrum is not common in the optimization area. The most related notion is perhaps "block Lipschitz constant" [8] for studying block coordinate descent (BCD) type methods, but it was not linked to the performance of SGD or Adam before. To our knowledge, we are not aware of any theory of Adam or GD built on the block diagonal structures or the blockwise spectra of Hessian. We now make an initial attempt in this direction. We first present the lower bound for GD.

**Proposition 1.** *(Lower bound for GD.) Consider $\min_w \mathcal{L}(w) = \frac{1}{2}w^T H w - h^T w$ where $H \in \mathbb{R}^{d \times d}$ is positive definite and $h \in \mathbb{R}^d$. Let $w_{GD}^t$ be the output of GD after $t$ steps. There exists a block diagonal matrix $H$, $h$ and an initial point $w^0$, s.t., for any $\eta$, we have:*

$$\mathcal{L}(w_{GD}^{t+1}) - \mathcal{L}^* \geq \left(1 - \frac{2}{\kappa + 1}\right)\left(\mathcal{L}(w_{GD}^t) - \mathcal{L}^*\right) \tag{5}$$

*where $\kappa$ is the condition number of $H$.*

Proposition 1 shows that GD has complexity $\tilde{\mathcal{O}}(\kappa)$ and such complexity is tight. Now we prove that Adam can achieves better complexity. This is because it chooses different learning rates for different block sub-matrix $H_l$ via its diagonal preconditinoner $D_{Adam}^0$. We consider generic random initialization that covers commonly used distributions such as Gaussian, Uniform, etc.

**Assumption 1.** *(Random initialization.) Assume the initialization $w^0$ is sampled from a continuous distribution, i.e., the probability measure (induced by $w^0$) of any zero-Lebesgue-measure set is 0.*

**Theorem 1.** *(Upper bound for Adam with $\beta_2 = 1$.) Consider the same setting as Proposition 1 and consider Adam with $\beta_1 = 0$ and $\beta_2 = 1$ as in (3). Assume the initialization satisfies Assumption 1. Let $w_{Adam}^t$ be the output of Adam after $t$ steps. Let $\eta = \min_{l \in [L]} \frac{1}{C_{l,1}}$. Then w.p.1., we have*

$$\mathcal{L}(w_{Adam}^{t+1}) - \mathcal{L}^* \leq \max_{l \in [L]}\left(1 - \frac{1}{\kappa_{Adam,l}}\right)\left(\mathcal{L}(w_{Adam}^t) - \mathcal{L}^*\right) \tag{6}$$

*where $\kappa_{Adam,l} = r\kappa_l$, $\kappa_l$ is the condition number of $H_l$, constant $r$ relates to $w^0$ defined as:*

$$r = \frac{\max_{l \in [L]} C_{l,2}^2}{\min_{l \in [L]} C_{l,1}^2}, \text{ where } C_{l,1} = \min_{i \in [d_l]} \frac{|[\nabla \mathcal{L}(w^0)]_{l,i}|}{\lambda_{l,1}}, C_{l,2} = \max_{i \in [d_l]} \frac{|[\nabla \mathcal{L}(w^0)]_{l,i}|}{\lambda_{l,1}}. \tag{7}$$

The proofs of the above theorems are shown in Appendix E. Theorem 1 states that Adam (with $\beta_2 = 1$) has complexity $\tilde{O}\left(r \cdot \max_{l \in [L]} \kappa_l\right)$. We note that coefficient $r$ depends on the ratio between initial gradient and the principal eigenvalue for each block, and smaller ratio would give faster convergence. We further remark that condition $\beta_2 = 1$ is necessary because any $\beta_2 < 1$ causes non-convergence issue [10, 21]. We restate their results in Proposition 2. The non-convergence is also observed in Figure 8 (c), where we find that the iterates of Adam quickly converge to near-optimal solutions, and then bounce back. As such, $\beta_2 = 1$ is necessary for asymptotic analysis. The analysis for $\beta_2 = 1$ is still meaningful since it still shows the effect of Adam's preconditioner.

As shown in [21], the non-convergence is due to the constant learning rate. Reducing the learning rate reduces the gap between $\mathcal{L}(w_{Adam}^t)$ and $\mathcal{L}^*$, but does not remove it.

**Proposition 2.** *(Non-convergence of constant-learning-rate Adam with $\beta_2 < 1$.) [21, Proposition 12, Figure 1] Consider $\min_{w \in \mathbb{R}} \mathcal{L}(w) = \frac{1}{2}w^2$. Consider Adam with $\beta_1 = 0$ and $\beta_2 < 1$ as in (4). Let $w_{Adam}^t$ be the output of Adam after $t$ steps. There exists a discrete limit cycle for (4) and $\liminf_{t \to \infty}\left(\mathcal{L}(w_{Adam}^t) - \mathcal{L}^*\right) > 0$.*

We now compare the complexity of Adam and that of GD. By Theorem 1, Adam is faster than GD when $r \cdot \max_{l \in [L]} \kappa_l \leq \kappa$. In the quadratic model with heterogeneous blocks (**Case 3**), our simulation over 1000 trials shows that $r \leq 1000$ with probability $\geq \frac{2}{3}$ when using standard Gaussian random initialization. Since $\max_{l \in [L]} \kappa_l \approx 1$, we have $r \cdot \max_{l \in [L]} \kappa_l \leq 1000$, w.h.p., and is about $5\times$ smaller than $\kappa = 5000$. So Adam could be $5\times$ faster than GD, w.h.p.. This is indeed observed in Figure 8 where Adam outperforms GD by a significant margin. We summarize the complexity of GD and Adam in Table 2.

**Remark: some common misconceptions.** During the review process, we find that readers might conclude that "Theorem 1 implies Adam under homogeneity has worse complexity than Adam under heterogeneity". We now clarify that this claim is *not* correct, and there is no conclusion on "whether Adam under homogeneity is faster or slower than Adam under heterogeneity". Similarly, Theorem 1 does *not* imply "Adam always converges similarly as GD under homogeneity". Though it is observed on CNNs, there is no general conclusion of this kind. For interested readers, we provide a detailed explanation in Appendix B.

Table 2: The complexity of GD and Adam for minimizing a strongly convex quadratic function with block diagonal Hessian. The symbol ✗ means non-convergence. $\kappa$ and $\kappa_l$ denote the condition number of the full Hessian and the block submatrix, respectively. $r$ is defined in (7).

| Optimizer | GD | Adam with $\beta_1 = 0$ and $\beta_2 = 1$ (3) | Adam with $\beta_1 = 0$ and $\beta_2 < 1$ (4) |
|---|---|---|---|
| Complexity | $\tilde{O}(\kappa)$ | $\tilde{O}\left(r \cdot \max_{l \in [L]} \kappa_l\right)$ | ✗ |

**How to obtain a tighter complexity bound of Adam?** It is valid to ask whether the complexity upper bound in Theorem 1 can be tightened, e.g., improve the factor of $r$. We point out it would be difficult if there is no extra structure on $H_l$. A key technical step is to bound the condition number of the preconditioned matrix $\kappa\left((D_{Adam,l}^0)^{-1} H_l\right)$. Intuitively, a diagonal preconditioner of $H_l$ is powerful when $H_l$ itself has a near-diagonal structure, e.g., pure diagonal, tridiagonal or diagonal dominant [30]. Unfortunately, it is unclear whether these structures hold in Transformers. Without any assumption on $H_l$, we find that the diagonal preconditioner of $D_{Adam}^0$ could *increase* the condition number. For instance, when using standard Gaussian initialization, in **case 3**, we find $\kappa\left((D_{Adam,l}^0)^{-1} H_l\right)$ equals $7.09\kappa_1$, $18.98\kappa_2$, $18.76\kappa_3$ for the 3 blocks, respectively (all averaged over 1000 trials). It would be interesting to explore if there are special structures of $H_l$ in Transformers such that Adam preconditioner can reduce $\kappa_l$, rather than increase it. We leave it as a future direction.

**More discussions on the theoretical advantage of Adam.** Although Adam preconditioner might not always reduce the "local" condition number $\kappa_l$, the coefficient in the complexity is now **independent of the "global" condition number** $\kappa$. As argued above, such changes in coefficient could lead to considerable improvement over GD. Such improvement in complexity is attributed to the block diagonal structure in Hessian as well as its heterogeneous blockwise spectrum. To our knowledge, such improvement is not shown in the existing literature. One possible reason is that: for the optimization community, it is very rare to analyze (near-) block-diagonal Hessian structure since typical problems do not have such structure. For instance, in the classical non-linear programming dataset [48], all problems have non-block-diagonal Hessian. We suggest a different perspective to characterize modern optimization problems. We believe our perspective is new because it is built upon multiple non-trivial findings.

In summary, our theory indicates that: for problems with block heterogeneity, the single-learning rate methods like GD can largely lag behind coordinate-wise learning rate methods like Adam.

## 5 Conclusion

In this work, we explore why SGD largely underperforms Adam on Transformers. we establish a phenomenon called block heterogeneity in Hessian and link it to the performance gap between Adam and SGD. We numerically verify our claim on various Transformers, CNNs, MLPs, and quadratic problems. Initial theory is also provided to support the claim.

## Acknowledgements

Yushun Zhang would like to thank Yinyu Ye, Wentao Ding, Guiyu Hong, Yingru Li, and Bohan Wang for the valuable discussions. The work of Ruoyu Sun was supported by NSFC (No. 12326608); Hetao Shenzhen-Hong Kong Science and Technology Innovation Cooperation Zone Project (No. HZQSWS-KCCYB-2024016), together with Tian Ding; University Development Fund UDF01001491, the Chinese University of Hong Kong, Shenzhen; Guangdong Provincial Key Laboratory of Mathematical Foundations for Artificial Intelligence (2023B1212010001). The work of Z.-Q. Luo was supported by the Guangdong Major Project of Basic and Applied Basic Research (No.2023B0303000001), the Guangdong Provincial Key Laboratory of Big Data Computing, and the National Key Research and Development Project under grant 2022YFA1003900.

## Broader Impacts

We explore why SGD performs worse than Adam for training Transformers. Our work can help the community better understand large AI model training. However, it would be a potential threat if the AI models are used for illegal usage.

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

# A Related Works

**On the unsatisfactory performance of SGD on Transformers**    There is an active line of works that explores why SGD performs significantly worse than Adam on Transformers. One representative hypothesis is that SGD cannot handle the heavy-tailed stochastic noise in language tasks [105]. However, Chen et al. [14], Kunstner et al. [46] reported that the gap between Adam and SGD maintains even in the full-batch case with no stochasticity, so there might be other reasons. Further, SGD performs worse than Adam on Vision Transformers on ImageNet (See Figure 10. Also see [92] for more evidence), so the data modality (e.g., language or vision tasks) might not be as crucial as the architecture. [104] showed that NLP tasks have "unbounded smoothness" issue and SGD with gradient clipping performs better than SGD in this case. Although clipping is an effective trick, we still observe a huge gap between clipped SGD and Adam [5], so there might be other reasons. Different from these works, we find SGD underperforms Adam because it uses one single learning rate for all blocks, which cannot handle the Hessian heterogeneity among blocks.

**Understanding of Adam.**    There was once a long-standing debate on the possible divergence of Adam [72]. The convergence for the unmodified versions is later established in [77, 107] for RMSprop and Adam. More convergence analyses of general adaptive gradient methods are listed later in this section. We here focus on the literature that explores the benefit of Adam. Xie et al. [93] show that Adam can help avoid saddle points, which is an orthogonal direction to this work. Wang et al. [84], Crawshaw et al. [19], Li et al. [51] show that Adam and its variant outperform SGD under relaxed smoothness conditions, based on the intuition that Adam can adaptively change its learning rate along iteration (over time). We pointed out that the theory is not complete: even for quadratic functions where the smoothness is fixed, SGD sometimes performs largely worse than Adam (Figure 8). This indicates that the benefit of Adam is not merely due to its ability to adaptively change the learning rate (over time), and there are other reasons for Adam's success. We show that an important benefit of Adam is its ability to handle the heterogeneity across blocks (over space).

Recent works [9, 91, 46, 54, 3] build a relation between Adam and the sign-based methods. Wu et al. [91] further showed that sign-based methods can be effective when the Hessian is diagonal and satisfies several other properties. However, as put by the authors, it seems "unclear to what extent these properties hold for real problems". Pan and Li [66] numerically found that the Adam can reduce the directional sharpness along trajectories, while its relation to fast convergence remains mysterious. A recent work [43] point out that Adam biases the trajectories towards regions where Hessian has "uniform diagonal entries" while SGD cannot. The distribution of Hessian diagonal entries is also investigated in [54]. The theory in [43] implies that Adam is faster when the Hessian is diagonal. However, as argued above, it is unclear whether the diagonal Hessian structure commonly holds in real problems. In fact, we find the Hessian is closer to a block-diagonal (instead of pure diagonal) structure on some small Transformers. In these cases, blockwise eigenvalues carry more information than diagonal entries, providing extra details such as the location of eigenvalues. We find that these extra details are important for distinguishing Adam and SGD.

**Hessian Spectrum Analysis.**    There are several important attempts to explore the Hessian spectrum of MLPs and CNNs. Early works [74, 75, 12] found that the Hessian spectra of MLPs and CNNs consist of a "bulk" together with a few "outliers". Papyan [69], Wu et al. [91], Liao and Mahoney [52] further characterized the bulks and outliers in theory. Papyan [67, 68] numerically built the relation between these "outliers" and the Gauss-Newton matrix. Sankar et al. [76] numerically explored the relation between Hessian of CNNs and Gauss-Newton matrix in each layer. They further found that most CNN layers contribute similarly to the overall loss surface. We find that this result is restricted to CNNs and does not hold on Transformers due to the heterogeneity. Gur-Ari et al. [39] showed that for MLPs and CNNs, gradient descent converges to a small subspace spanned by a few top eigenvectors of the Hessian. Yao et al. [97], Zhang et al. [103] explored the relation between the Hessian spectrum of CNNs and some training phenomena such as the effect of batch sizes. Ghorbani et al. [32], Yao et al. [98] focused on explaining the effectiveness of techniques such as BatchNorm. Note that all these works are restricted to MLPs and CNNs, while we study the Hessian of Transformers (in addition to CNNs and MLPs) as well as its impacts on different optimizers.

---

[5]For all NLP tasks, clipping is performed immediately after backpropagation. So in Figure 10, SGD in NLP tasks essentially refers to clipped SGD.

**On the difficulties of Transformer training.** Transformers are known to be difficult to train. Researchers have attributed the training difficulties to various phenomena in different components of Transformers, including: the logits divergence or the rank degeneracy in the outputs of attention layers [26, 65, 90, 101, 24, 17]; the growth of parameter norm in attention layers [61]; over-reliance on residue branches [56]; and some negative impact of layer norm [15, 102, 41]. These phenomena have a strong correlation with gradient vanishing or explosion in Transformers [102, 56, 41, 94, 65, 87, 90, 62], which leads to training difficulties.

Several solutions have been proposed. Liu et al. [56] numerically observed that adaptive gradient methods can (partly) overcome gradient vanishing by giving "consistent update magnitude", while it seems unclear how consistent update magnitude would help optimization in principle. Researchers further develop training tricks such as warmup learning rate [55, 94], temperature scaling [65], better initialization [102, 41, 87, 5, 96], and variants of Layer Norm [64, 88, 94, 87, 24]. Recent researchers also suggest using z-loss regularization [17, 95] and tuning hyperparameters of Adam [107, 90]. All these tricks can help mitigate gradient explosion or vanishing. Nevertheless, training large-scale Transformers remains challenging [106, 100, 90, 62, 17]. Different from all aforementioned works, we investigate the training difficulties of Transformers through the eigenvalues of Hessian. We establish a strong correlation between "the blockwise Hessian spectra of Transformers" and "why SGD largely underperforms Adam on Transformers". We realize that our attempt is just a first step towards understanding Transformer training, and we believe there is rich information hidden in Hessian and we leave more fine-grained analysis as future works.

**Convergence analysis of general adaptive gradient methods** There is extensive convergence analysis for adaptive gradient methods. For instance, researchers study the convergence of AMSGrad [72, 108], RMSprop [99], AdaFom [16], AdaBound [58], and Adam with iterate-dependent hyper-parameters [109, 13, 31]. The convergence of Adam is also explored in [23, 85]. There is also an active line of theoretical research on the convergence of AdaGrad [28], we recommend [86] for more detailed introduction. In this work, we do not focus on the convergence analysis. Rather, we explore the quantitative difference between the loss landscape of CNNs and Transformers and how it impact the behaviors of SGD and Adam.

# B    More Results and Discussions

**Performance comparison of AdamW and SGD on different Architectures.**    Here, we show the performance comparison of AdamW and SGD on different models. All the vision models are trained on ImageNet. Language models are trained on different English corpus. We grid-search the learning rates for SGD and Adam under the same budget and report the best result for each optimizer. See Appendix D.1 for more implementation details.

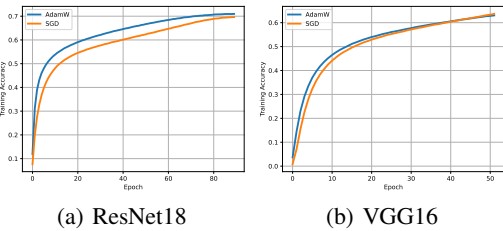

(a) ResNet18                    (b) VGG16

Figure 9: Performance of AdamW and SGD on CNNs including ResNet18 and VGG16. SGD and Adam perform similarly on these tasks.

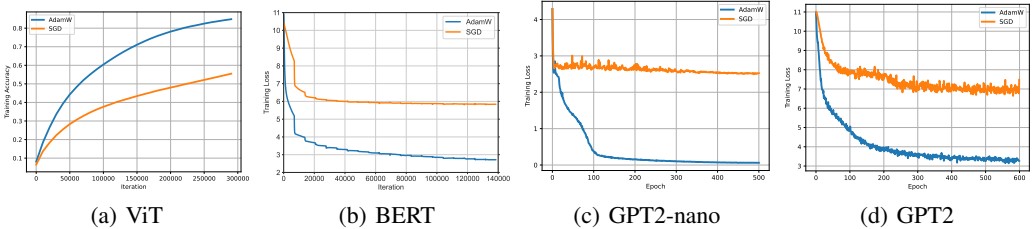

(a) ViT                 (b) BERT                 (c) GPT2-nano                 (d) GPT2

Figure 10: Performance of AdamW and SGD on Transformers including ViT, BERT, GPT2-nano, and GPT2. SGD performs significantly worse than Adam on these tasks.

**More results for SGD under careful tuning.**    For ViT-base, GPT2-nano, and BERT, we further present the performance of SGD under learning rate grid search. For ViT-base training on ImageNet, we report the results after 30 epochs (or equivalently, about 30k iterations). We cannot afford further training ViT-base due to the limited hardware resources (a complete run of 90 epochs would take > 2 weeks for each curve). As shown in Figure 11, SGD consistently performs worse than Adam on ViT-base, GPT2, and BERT. The best results for SGD are picked out and presented in Figure 10.

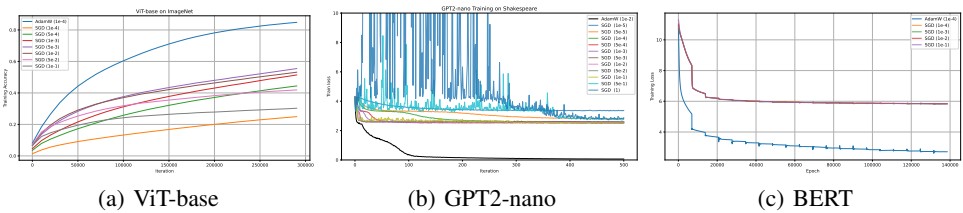

(a) ViT-base                     (b) GPT2-nano                     (c) BERT

Figure 11: Performance of SGD under careful tuning. On ViT-base, GPT2-nano, and BERT, we carefully tune the learning rate of SGD and show all the results here. For all these Transformer tasks, SGD is still significantly worse than AdamW even after careful tuning.

Note that we are not the first ones to report that "SGD performs worse than Adam on ViT". An influential work [92] also reports that SGD is worse than Adam on vanilla ViT. The authors report that "SGD yields significantly worse results than AdamW (on ViT)", and "ViT often fails to converge with SGD " (their Figure 3). These results align with our findings in Figure 11.

**Detailed training curves of Figure 5.** In Figure 12, we present the detailed training curves for the man-made MLP in Figure 5. We find that SGD performs worse as heterogeneity grows, while Adam still performs well.

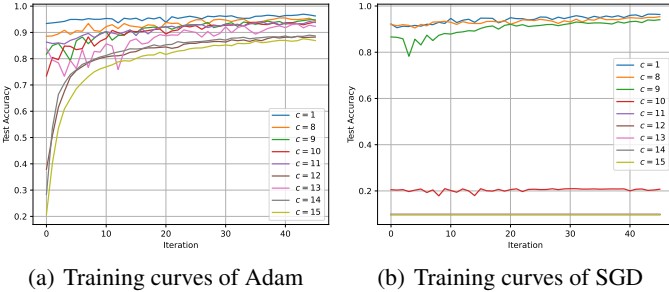

(a) Training curves of Adam      (b) Training curves of SGD

Figure 12: The training curves of SGD and Adam on MNIST with 4-layer MLPs under different degrees of block heterogeneity $c$. We observe that SGD performs worse as heterogeneity grows, while Adam remains unaffected.

**The evolution of Hessian heterogeneity along training.** For ViT-base, we further investigate the evolution of block heterogeneity of Hessian along the training. As shown in Figure 13, we find that heterogeneity attenuates along the training. We further take the checkpoint of ViT-base at 50% training step and switch AdamW to SGD, we observe that now SGD performs better than training from scratch as in Figure 10 (a). SGD performs better here because there is less heterogeneity when initializing in the middle of training.

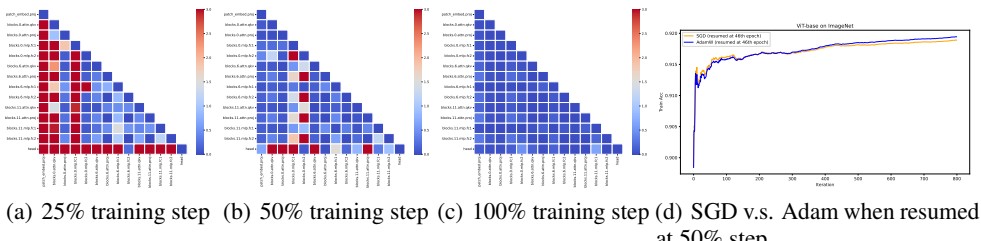

(a) 25% training step  (b) 50% training step  (c) 100% training step  (d) SGD v.s. Adam when resumed at 50% step

Figure 13: For ViT-base, we plot the evolution of heterogeneity of Hessian along training. We find that heterogeneity attenuates along training. (d): we take the checkpoint of ViT-base at 50% training step and switch AdamW to SGD, we find that now SGD performs better than training from scratch as in Figure 10 (a).

**Simplifed SLQ for calculating $JS^0$ in Section 3.4.** We note that $JS^0$ in Table 1 is rather expensive to compute due to the computational overhead of SLQ: it requires comparable time to one training run. Fortunately, we find the original SLQ is redundant for measuring hessian heterogeneity. We propose the following simple tweaks to significantly reduce the computation time, while still effectively detecting the Hessian heterogeneity. We call it simplified SLQ.

- Change the hyperparemters of SLQ, including:
  - We change $\texttt{num}_v = 10$ to $\texttt{num}_v = 1$. In SLQ, $\texttt{num}_v$ decides the number of random Gaussian vectors to approximate the expected quadrature. It is reasonable to reduce $\texttt{num}_v$ because in high dimensional space, random vectors tend to concentrate around their mean, so one random sample can already be informative enough.
  - We change the Lanzcos step $m = 100$ to $m = 10$. The reduction on Lanzcos step will have a coarse estimation on the middle eigenvalue, but won't affect much the heterogeneity, which is more dependent on the extreme eigenvalues.
- Randomly sample a subset of blocks and reduce batch size for estimating the spectrum. We uniformly sample 50% blocks and choose batch size = 32 (previously batch size = 1024).

We report the result and runtime in Table 3. As a result, the simplified SLQ can obtain the same message as the original SLQ: $JS^0$ of ResNet is about 100x smaller than that of BERT. Further, the simplified SLQ is highly efficient to compute. With this simplified SLQ, we believe our method can efficiently scale to larger models. The result is tested on a single V100.

Table 3: $JS^0$ computed by simplified SLQ. We find that the simplified SLQ can obtain the same message as the original SLQ: $JS^0$ of ResNet is about 100x smaller than that of BERT. Further, the simplified SLQ is efficient to compute.

| Model | JSO | Time for JSO | Time for Training | Time for JSO / Time for training |
|-------|-----|--------------|-------------------|----------------------------------|
| BERT | 98.8344 | 20 s | 4 h | 0.0014 |
| ResNet18 | 0.3569 | 65 s | 87.5 h | 0.0002 |

**Blockwise spectra for quadratic models in Section 4.1.**    We here visualize the blockwise spectrum for the quadratic models in **Case 1** and **Case 2**. These spectra are collected from GPT2 and ResNet18, respectively.

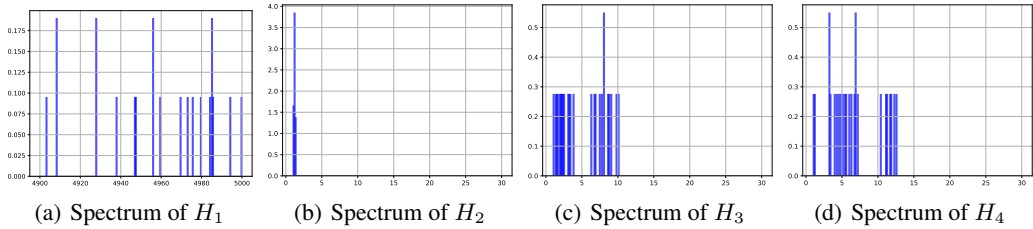

(a) Spectrum of $H_1$    (b) Spectrum of $H_2$    (c) Spectrum of $H_3$    (d) Spectrum of $H_4$

Figure 14: Histogram of eigenvalues of each block in **Case 1** (the heterogeneous case). The eigenvalues in the four blocks are sampled from the spectrum of the embedding layer; 3rd Query, 3rd Value, 3rd MLP (`fc` layer) in GPT2, respectively. All the eigenvalues are shifted and proportionally scaled such that: the objective function is strong convex; the condition number of Hessian equals 5000; their relative ranges are preserved; and the block heterogeneity is preserved.

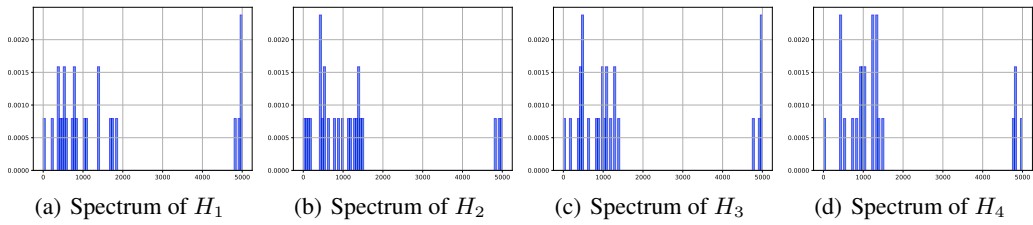

(a) Spectrum of $H_1$    (b) Spectrum of $H_2$    (c) Spectrum of $H_3$    (d) Spectrum of $H_4$

Figure 15: Histogram of eigenvalues of each block in **Case 2** (the homogeneous case). The eigenvalues in the four blocks are sampled from the spectrum of 1st to 4th convolution layers in ResNet18, respectively. All the eigenvalues are shifted and proportionally scaled such that: the objective function is strong convex; the condition number of Hessian equals 5000; their relative ranges are preserved; and the block homogeneity is preserved.

**Does our theory imply that Adam under homogeneity is slower than Adam under heterogeneity?**
[6] During the review process, we find that readers might conclude that "Theorem 1 implies Adam under homogeneity has worse complexity than Adam under heterogeneity". We would like to clarify that this conclusion is *not* correct, and there is no conclusion on "whether Adam under homogeneity is faster or slower than Adam under heterogeneity ". We explain as follows.

Our theoretical result states that Adam has complexity $\mathcal{O}\left(\max_l \kappa_l\right)$. If our result implies the above conclusion, one needs the following argument: when changing heterogeneity to homogeneity, $\max_l \kappa_l$

---

[6]We would like to thank the anonymous Reviewer pTmy for raising this insightful question.

increases, and thus Adam is slower. However, "changing heterogeneity to homogeneity" does not necessarily mean "$\max_l \kappa_l$ increases". Actually, $\max_l \kappa_l$ can change in an arbitrary way (can increase, decrease, or keep the same) when changing the heterogeneity. We provide three examples below.

We will use Adam (homo) to denote the convergence rate of Adam on homogeneous Hessian, similarly for Adam (hetero).

**Example 1: Adam (homo) is same as Adam (hetero).**

- Case 1-1 (homogeneous): eigenvalues are {1,2}, {1,2}
- Case 1-2 (heterogeneous): eigenvalues are {1,2}, {11,12}

Since $\max_l \kappa_l$ are the same for both Case 1-1 and 1-2, Adam (homo) is the same as Adam (hetero).

**Example 2: Adam (homo) is faster than Adam (hetero).**

- Case 2-1 (homogeneous): eigenvalues are {1,1.5}, {1,1.5}
- Case 2-2 (heterogeneous): eigenvalues are {1,2}, {11,12}

Since Case 2-1 has smaller $\max_l \kappa_l$ than Case 2-2, Adam (homo) is faster than Adam (hetero).

**Example 3: Adam (homo) is slower than Adam (hetero).**

- Case 3-1 (homogeneous): eigenvalues are {1,12}, {1,12}
- Case 3-2 (heterogeneous): eigenvalues are {1,2}, {11,12}

Since Case 3-1 has larger $\max_l \kappa_l$ than Case 3-2, Adam (homo) is slower than Adam (hetero).

To sum up, there is no conclusion on "whether Adam under homogeneity is faster or slower than Adam under heterogeneity ". Either case can happen.

One possible source of confusion may come from the numerical examples (**Case 3 and 4** in Section Section 4.1). If comparing two figures, Adam (homo) in **Case 3** is slower than Adam (hetero) in **Case 4**. But as argued above, this is just one example, and it does *not* show Adam (homo) is always slower than Adam (hetero).

## C  More Preliminaries

### C.1  Preliminaries on Optimizers

Here we provide a detailed description of the optimizers mentioned in the full script. We consider the minimizing $\mathcal{L}(w) \equiv \frac{1}{n} \sum_{i=1}^{n} \mathcal{L}_i(w)$, where $n$ is the number of minibatches, $\mathcal{L}_i(w)$ is the loss of $i$-th minibatch and $w \in \mathbb{R}^d$ is the neural network parameters. We denote the gradient of the training loss w.r.t. neural network parameters as $\nabla \mathcal{L}(w) \in \mathbb{R}^d$. We use $\nabla \mathcal{L}_i(w) \in \mathbb{R}^d$ to denote the $i$-th minibatch counterparts. We use $w^t$ to denote the variable at the $t$-th step. In Algorithm 2 and 3, $\circ$, division and square-root are elementwise operations. In the line 7 and 8 of Algorithm 2, $(\beta_1)^t$ and $(\beta_2)^t$ indicates the $t$-th power of $\beta_1, \beta_2$. In the PyTorch default setting, $(\beta_1, \beta_2, \epsilon) = (0.9, 0.999, 1e\text{-}8)$ for Adam and $\beta_1 = 0.9$ for SGD.

---

**Algorithm 1** Stochastic Gradient Descent with Momentum (SGD)

---

1: Initialize $w^0$ and choose $0 \leq \beta_1 < 1$ and $\eta_0 > 0$
2: **for** $t = 1 \to \infty$ **do**
3:     Uniformly sample $\tau^t$ from the index set $\{1, 2, \cdots, n\}$
4:     $m^t = \beta_1 m^t + \nabla \mathcal{L}_{\tau^t}(x^t)$
5:     $x^{t+1} = x^t - \eta_t m^t$
6: **end for**

---

**Algorithm 2** AdamW

---

1: Initialize $x^0$, $m^0 = v^0 = 0$, $0 \leq \beta_1 < 1$, $0 \leq \beta_2 < 1$, $\epsilon > 0$, $\eta^0 > 0$, and weight decay coefficient $\lambda$
2: **for** $t = 1 \to \infty$ **do**
3:     Uniformly sample $\tau^t$ from the index set $\{1, 2, \cdots, n\}$
4:     $w^{t+1} = w^t - \eta^t \lambda w^t$
5:     $m^t = \beta_1 m^t + (1 - \beta_1) \nabla \mathcal{L}_{\tau^t}(w^t)$
6:     $v^t = \beta_2 v^t + (1 - \beta_2) \nabla \mathcal{L}_{\tau^t}(w^t) \circ \nabla \mathcal{L}_{\tau^t}(w^t)$
7:     $\hat{m}^t = \frac{m^t}{1 - (\beta_1)^t}$
8:     $\hat{v}^t = \frac{v^t}{1 - (\beta_2)^t}$
9:     $w^{t+1} = w^{t+1} - \eta_t \frac{\hat{m}^t}{\sqrt{\hat{v}^t} + \epsilon}$
10: **end for**

---

**Algorithm 3** Adam with no bias correction

---

1: Initialize $x^0$, $m^0 = \nabla \mathcal{L}_{\tau^t}(w^0)$, $v^0 = \nabla \mathcal{L}_{\tau^t}(w^0) \circ \nabla \mathcal{L}_{\tau^t}(w^0)$, $0 \leq \beta_1 < 1$, $0 \leq \beta_2 < 1$, $\epsilon > 0$, $\eta^0 > 0$
2: **for** $t = 1 \to \infty$ **do**
3:     Uniformly sample $\tau^t$ from the index set $\{1, 2, \cdots, n\}$
4:     $m^t = \beta_1 m^t + (1 - \beta_1) \nabla \mathcal{L}_{\tau^t}(w^t)$
5:     $v^t = \beta_2 v^t + (1 - \beta_2) \nabla \mathcal{L}_{\tau^t}(w^t) \circ \nabla \mathcal{L}_{\tau^t}(w^t)$
6:     $w^{t+1} = w^{t+1} - \eta_t \frac{m^t}{\sqrt{v^t} + \epsilon}$
7: **end for**

---

### C.2  Preliminaries on the Stochastic Lanczos Quadrature Method

**Additional notations.** Given a real symmetric matrix $H \in \mathbb{R}^{d \times d}$, we denote $tr(H)$ as its trace and $Q^T \Lambda Q$ as its spectral decomposition, where $Q = [q_1, \ldots, q_d]$, $\Lambda = \text{diag}(\lambda_1, \ldots, \lambda_d)$ and $\lambda_1 \geq \lambda_2 \cdots \geq \lambda_d$. We denote the condition number of $H$ as $\kappa = \lambda_1 / \lambda_d$. We define matrix function as $f(H) := Q^T f(\Lambda) Q$, where $f(\Lambda) = \text{diag}(f(\lambda_1), \ldots f(\lambda_d)) \in \mathbb{R}^{d \times d}$. We use $\mathbb{N}$ to denote the set of positive integers. We use $\| \cdot \|_2$ to denote the Euclidean norm.

Approximation of the Hessian spectrum can be formulated as a trace estimation problem, as introduced in [53, 82]. First, the spectrum (eigenvalue density) of Hessian $H$ can written as:

$\phi(t) = \frac{1}{d}\sum_{i=1}^{d} \delta(t - \lambda_i)$, where $\lambda_i$ are the eigenvalues of $H$ and $\delta$ is the Dirac $\delta$-function. Then, we replace the delta functions by a Gaussian blurring function: $\phi(t) \approx g(t) := \frac{1}{d}\sum_{i=1}^{d} f(\lambda_i)$, where $f(\lambda) := \frac{1}{\sigma\sqrt{2\pi}}\exp\left(-\frac{(t-\lambda)^2}{2\sigma^2}\right)$. By definition of matrix function, it is easy to see that $g(t) = \frac{1}{d}tr(f(H))$. As such, spectrum approximation could be formulated as a trace estimation problem, i.e., estimating $\frac{1}{d}\operatorname{tr}(f(H))$, where $H \in \mathbb{R}^{d\times d}$ is a real symmetric matrix.

Trace estimation problems could be solved efficiently by the Stochastic Lanczos Quadrature Method (SLQ) [35]. For the ease of readers, we re-organize and summarize the existing literature ([35, 82, 32]) and provide a detailed description of SLQ in our context. SLQ consists of the following steps.

**Step 1.** We Approximate the trace of matrix function as $\frac{1}{d}tr(f(H)) = \mathbb{E}(v^T f(H)v) \approx \frac{1}{n_v}\sum_{i}^{n_v} v_i^T f(H)v_i$, where $v = u/\|u\|_2$ and $u$ is a Rademacher random vector (each entry of $u$ independently takes $\pm 1$ with probability $1/2$). This step is called Huchinson's estimation [42].

Note that we can also replace the Rademacher random vector $u$ by a unit Gaussian vector (i.e., $u \sim N(0, I_{d\times d})$) and the unbiasedness still holds [4]. In our implementation, we sample $u \sim N(0, I_{d\times d})$ because there is an efficient built-in PyTorch function for generating Gaussian vectors.

SLQ estimates $v_i^T f(H)v_i$ for $i \in [n_v]$ and then take the average. To understand SLQ, we only need to understand how it estimates each individual quadratic form. To simplify the notation regarding $i$, from now on, we will discuss how to estimate $v^T f(H)v$, where $v = u/\|u\|_2$ and $u$ is a unit Gaussian vector.

**Step 2-1.** We rewrite $v^T f(H)v$ as a Riemann-Stieltjes integral [34]:

$$v^T f(A)v = \sum_{i=1}^{d} \left(v^T q_i\right)^2 f(\lambda_i) = \int_{\lambda_d}^{\lambda_1} f(\lambda)d\mu(\lambda), \tag{8}$$

where $\mu$ is a measure on $(\mathbb{R}, \mathbb{B})$ defined as follows ($\mu(\lambda)$ denotes the measure of set $\{x; x \leq \lambda\}$):

$$\mu(\lambda) = \begin{cases} 0 & \lambda < \lambda_d \\ \sum_{i=1}^{k}\left(v^T q_i\right)^2 & \lambda_k \leq \lambda < \lambda_{k+1} \\ \sum_{i=1}^{d}\left(v^T q_i\right)^2 & \lambda \geq \lambda_1 \end{cases}. \tag{9}$$

**Step 2-2.** Unfortunately, this integral is difficult to compute. This is because the measure $\mu$ are related to the eigen-pairs of $H$, which are unknown. It seems unclear how to directly integrate over an unknown measure. As such, we further approximate this integral by a computationally friendly quantity, such as:

$$\int_{\lambda_d}^{\lambda_1} f(\lambda)d\mu(\lambda) \approx \sum_{j=1}^{m} c_j f(x_j). \tag{10}$$

We hope to design $\{(c_j, x_j)\}_{j=1}^{m}$ with a reasonable number of $m$ such that the estimation error is small. Fortunately, the Gaussian Quadrature method provides a generic design principle of $\{(c_j, x_j)\}_{j=1}^{m}$ [34, 29]. It is proved that: when $f(\lambda)$ is not "too complicated" (e.g. $f(\lambda)$ is a polynomial), then there exists $\{(c_j, x_j)\}_{j=1}^{m}$ which gives a high quality estimation of integral (8). The required number of $m$ is related to "how complicated the $f(\lambda)$ is". Such $\{(c_j, x_j)\}_{j=1}^{m}$ are called the Gaussian Quadrature rules. $c_j$ and $x_j$ are called the "weights" and the "nodes" of the Gaussian Quadrature rules. A representative theorem is as follows: when $f(\lambda)$ is a polynomial with degree $< 2m$, then the Gaussian Quadrature rules give the exact approximation of integral (8).

**Theorem 2.** *[Rewrited based on [89]] Suppose we have a sequence of orthogonal polynomials* $\{p_k(x)\}_{k=1}^{m}$ *w.r.t. measure $\mu$, that is:* $\int_{\lambda_d}^{\lambda_1} p_n(x)p_m(x)d\mu(x) = \delta_{m,n}$, *where $\delta_{m,n} = 1$ if $m = n$ and $\delta_{m,n} = 0$, otherwise. Assume $f(x)$ is a polynomial with degree $< 2m$, then there exists* $\{(c_j, x_j)\}_{j=1}^{m}$ *s.t.* $\int_{\lambda_d}^{\lambda_1} f(\lambda)d\mu(\lambda) = \sum_{i=j}^{m} c_j f(x_j)$. *The equality holds when $x_j$ are the roots*

of $p_m(x)$ and $c_j = \int_{\lambda_d}^{\lambda_1} \prod_{j \neq i} \frac{x-x_i}{x_j-x_i} d\mu$. *Such choice of $\{(c_j, x_j)\}_{j=1}^m$ are called the Gaussian Quadrature rules.*

Theorem 2 shows the existence of good $\{(c_j, x_j)\}_{j=1}^m$ and their general form. In fact, it is also shown that Gaussian Quadrature is optimal: no other $\{(c_j, x_j)\}_{j=1}^m$ can achieve zero approximation error for higher degree polynomials $f(\lambda)$ [34]. However, it is often difficult to find these quadrature rules [36]. There are at least three questions in sequel:

- **1)** how to find the orthogonal polynomials $\{p_k(x)\}_{k=1}^m$ w.r.t. an unknown measure $\mu$?
- **2)** how to efficiently find the roots of $p_m(x)$, which gives the nodes $x_j$?
- **3)** how to efficiently calculate the weights $c_j = \int_{\lambda_d}^{\lambda_1} \prod_{j \neq i} \frac{x-x_i}{x_j-x_i} d\mu$?

We first answer question **2)** and **3)** and leave question **1)** for later discussion.

Now suppose that we have found the orthogonal polynomials $\{p_k(x)\}_{k=1}^m$ w.r.t. $\mu$. Recall that any orthogonal polynomial has the following "three-term" recursion [34]:

$$p_{k+1}(x) = (x - \alpha_{k+1}) p_k(x) - \beta_k p_{k-1}(x), k = 0, 1, \ldots,$$

where $p_{-1}(x) \equiv 0, p_0(x) \equiv 1, \alpha_{k+1} = \frac{\langle xp_k, p_k \rangle}{\langle p_k, p_k \rangle}$ and $\beta_k = \frac{\langle p_k, p_k \rangle}{\langle p_{k-1}, p_{k-1} \rangle}$. Define $P_m(x) = (p_0(x), p_1(x), \ldots p_{m-1}(x))^T \in \mathbb{R}^m$, we can rewrite the recursion formula in matrix form (given $x$): $xP_m = J_m P_m + \beta_m p_m(x) e^m$, where $e^m$ is the last column of identity matrix $I_{m,m}$ and $J_m$ is called Jacobi matrix of order $m$:

$$J_m = \begin{pmatrix} \alpha_1 & \sqrt{\beta_1} & & & \\ \sqrt{\beta_1} & \alpha_2 & \sqrt{\beta_2} & & \\ & \sqrt{\beta_2} & \alpha_3 & \sqrt{\beta_3} & \\ & & \ddots & \ddots & \ddots \end{pmatrix} \in \mathbb{R}^{m \times m}$$

It turns out that $J_m$ can help us find the Gaussian Quadrature rules $\{(c_j, x_j)\}_{j=1}^m$ and thus provide answers for question **2)** and **3)**. This is shown in the following theorem.

**Theorem 3.** *[34] For the Gaussian Quadrature, $\{x_j\}_{j=1}^m$ are the eigenvalues of $J_m$ and $\{c_j\}_{j=1}^m$ are the squares of the first elements of the normalized eigenvectors of $J_m$.*

The proof of Theorem 3 is based on Christoffel-Darboux relation [11]. Now, the remaining question is: how to find the Jacobian matrix $J_m$ of a sequence of orthogonal polynomials w.r.t. an unknown measure $\mu$? Note that we no longer need to answer question **1)** if $J_m$ is found, since $J_m$ is sufficient for us to find the Gaussian quadrature rules. However, it seems impossible to find $J_m$ if no information of $\mu$ is provided. The good news is: when the $\mu$ is specified as in (9), there exists an efficient way to find $J_m$.

**Step 3.** When $\mu$ is specified as in (9), $J_m$ can be exactly found in $m$ steps using the Lanczos algorithm [47], as shown in Algorithm 4. This method takes a real symmetric matrix as input and returns a tridiagonal matrix. It was originally proposed to solve eigenvalue problems. Later, researchers found a deep connection between the Lanczos algorithm and orthogonal polynomials, which further connects this method to the Gaussian quadrature. The method (of finding the Gaussian quadrature by the Lanczos algorithm) is called the Lanczos quadrature [35, 6, 34]. An extremely elegant but highly nontrivial result is as follows:

**Theorem 4.** *[34] Given a real symmetric matrix $H \in \mathbb{R}^{d \times d}$ and an arbitrary vector $v \in \mathbb{R}^d$ with unit Euclidean norm, we define the measure $\mu$ as in (9) based on this $H$ and $v$. Then $m$ steps of the Lanczos algorithm return the Jacobian matrix $J_m$ of orthogonal polynomials w.r.t. to $\mu$.*

After $J_m$ is found by the Lanczos algorithm, we perform spectral decomposition of $J_m \in \mathbb{R}^{m \times m}$ to get its eigen-pairs. Using Theorem 3, we successfully get the Gaussian quadrature rules and thus we can approximate the quadratic form $v^T f(H)v$. By averaging over different random vectors $v$ we can then approximate $\frac{1}{d} tr(f(H))$. This concludes the derivation of SLQ for the trace estimation problem.

The full procedure of SLQ is shown in Algorithm 5. We note that SLQ is efficient in theory. Ubaru et al. [82] show that SLQ converges faster than any other polynomial expansion method for spectrum estimation (e.g., Chebyshev methods used in [2]). See [82, Theorem 4.1] for a formal statement.

We remark that there are at least four versions of the Lanczos algorithm in **Step 3**. Here, we adopt the version in Algorithm 4 since it is known to be the most numerically stable version [20, 73, 89]. Throughout this work, we choose $f(\cdot)$ as the Gaussian blurring function $f(\lambda) := \frac{1}{\sigma\sqrt{2\pi}}\exp\left(-\frac{(t-\lambda)^2}{2\sigma^2}\right)$ for spectrum approximation. We plot the spectrum by sweeping $t$ from the minimal node to the maximal node in Gaussian Quadrature rules.

---

**Algorithm 4** The Lanczos Algorithm

---

1: Input a matrix-vector product $Hv_1 \in \mathbb{R}^d$, where $H$ is a real symmetric matrix and $v_1$ is an arbitrary vector with Euclidean norm 1. Choose $m \in \mathbb{N}$

2: **Initialization:** Let $w_1' = Hv_1$, $\alpha_1 = (w_1')^T v_1$, $w_1 = w_1' - \alpha_1 v_1$

3: **for** $j = 2 \to m$ **do**

4:     Let $\beta_j = \|w_{j-1}\|_2$ (also Euclidean norm)

5:     If $\beta_j \neq 0$, then let $v_j = w_{j-1}/\beta_j$,
       else pick as $v_j$ an arbitrary vector with Euclidean norm 1 that is orthogonal to all of $v_1, \ldots, v_{j-1}$

6:     Let $w_j' = Av_j$

7:     Let $\alpha_j = (w_j')^T v_j$

8:     Let $w_j = w_j' - \alpha_j v_j - \beta_j v_{j-1}$

9: **end for**

10: Let $V$ be the matrix with columns $v_1, \ldots, v_m$

11: Let $T = \begin{pmatrix} \alpha_1 & \beta_2 & & & & & 0 \\ \beta_2 & \alpha_2 & \beta_3 & & & & \\ & \beta_3 & \alpha_3 & \ddots & & & \\ & & \ddots & \ddots & \beta_{m-1} & & \\ & & & \beta_{m-1} & \alpha_{m-1} & \beta_m & \\ 0 & & & & \beta_m & \alpha_m \end{pmatrix}$

12: Return $T$

---

**Algorithm 5** The Stochastic Lanczos Quadrature Method

---

1: Choose $\text{num}_v, m \in \mathbb{N}$. Sample $\text{num}_v$ i.i.d. $v_i$ from normalized Rademacher distribution, $i \in [\text{num}_v]$

2: **for** $i = 1 \to \text{num}_v$ **do**

3:     Run $m$ steps of the Lanczos Algorithm 4 with input $Hv_i$, returns $T \in \mathbb{R}^{m \times m}$

4:     Compute eigenvalue decomposition $T = Q\Lambda Q^T$

5:     Compute the nodes $x_i = (\Lambda_{ii})_{i=1}^m$ and weights $c_i = (Q_{1,i}^2)_{i=1}^m$

6:     Return $q_i(t) = \sum_{i=1}^m c_i f\left(x_i; t, \sigma^2\right)$

7: **end for**

8: Return $\frac{1}{\text{num}_v}\sum_{i=1}^{\text{num}_v} f\left(\ell_i; t, \sigma^2\right)$

---

## D  More Eperimental Details

### D.1  Implementation Details on SLQ and Training Configurations

**Implementation and Running Time Analysis.**  We provide a simple PyTorch implementation of SLQ. The only query SLQ makes to the neural network is the Hessian vector product, which is attained using the auto-differentiation framework [70]. To assure the accuracy of the Lanczos algorithm, we remove all the randomness in the forward and backward passes, including: data shuffling order, data augmentation, and dropout, etc.. Since Flash Attention [22] does not support the calculation of Hessian-vector product, we implement all attention blocks in the naive way. For the calculation of the blockwise Hessian spectrum $\nabla^2 \mathcal{L}(w_l)$, we sample $u_l \sim N(0, I_{d_l \times d_l})$ and set $v_l = u_l / \|u_l\|_2 \in \mathbb{R}^{d_l}$. Then we run Algorithm 5 by taking $\nabla^2 \mathcal{L}(w_l)$ and $v_l$ as inputs. We choose the hyperparameters as $m = 100$ and $n_v = 10$ in all experiments. $\sigma$ is tuned based on visual effects. These hyperparameters are reported to reach highly accurate estimation with error $< 10^{-14}$ [32].

We now briefly discuss the computational cost of SLQ. The major computational expense of SLQ is the repeated Hessian-vector product operations in Lanczos algorithm in **Step 3**. Recall $\nabla^2 \mathcal{L}(w)d = \frac{1}{n} \sum_{i=1}^{n} \nabla^2 \mathcal{L}_i(w)d$, so each Hessian-vector product operation requires **(i)** calculating $\nabla^2 \mathcal{L}_i(w)d$; **(ii)** repeating **(i)** on all data. We point out that **(i)** can be computed efficiently and precisely with just two backpropagation passes [70]. The major computational bottleneck lies in **(ii)** due to the large $n$. Our largest-scale experiment for Hesian spectrum is GPT2 (125M) on Openwebtext, where the number of tokens $n = 9$ Billon. To calculate $\nabla^2 \mathcal{L}(w)d$ on all these 9B tokens, it requires about 9 GPU days on eight A100-80GB GPUs. Since SLQ requires at least 1,000 times query of $\nabla^2 \mathcal{L}(w)d$, a complete run of SLQ would take at least 9,000 days on eight A100-80GB GPUs, which is unaffordable. In this work, we use the largest possible batch size (with gradient accumulation tricks) to approximate $\nabla^2 \mathcal{L}(w)$ under the constraints of GPU bandwidth and time limit. More detailed setup of SLQ are shown as follows.

- **ResNet18 (18M) and VGG16 (138M) on ImageNet.** We use the code base of PyTorch Examples [7]. We use batch size $= 1024$. For the calculation of the blockwise Hessian spectra, we apply SLQ to all parameter blocks except for the BatchNorm layers. In total, it takes about 3 days on one V100 GPU to estimate all the blockwise Hessian spectra and the full Hessian spectrum.

- **ViT-base (86M) on ImageNet.** We use the code base of PyTorch Image Models [8]. We use batch size $= 1024$. Due to the large number of parameters, we are not able to calculate the blockwise Hessian spectra for all parameter blocks. Instead, we apply SLQ to: the embedding layer; the output layer; the 1-st, 6-th, 12-th attention blocks; and the 1-st, 6-th, 12-th MLP blocks (note that the 12-th attention and MLP blocks are the final ones). In total, it takes about 3 days on one V100 GPU to estimate all the blockwise Hessian spectra and the full Hessian spectrum.

- **BERT(40M) on Cornell Movie-Dialogs Corpus.** We use the code base from the blog [9]. We use batch size $= 327,680$ tokens. For the calculation of the blockwise Hessian spectra, we apply SLQ to all parameter blocks except for the LayerNorm layers. In total, it takes about 12 hours on one V100 GPU to estimate all the blockwise Hessian spectra and the full Hessian spectrum.

- **GPT2-nano (11M) on Shakespeare.** We use the code base of NanoGPT [10]. We use batch size $= 163,840$ tokens. For the calculation of the blockwise Hessian spectra, we apply SLQ to all parameter blocks with even indices, except for the LayerNorm layers. In total, it takes about 12 hours on one V100 GPU to estimate all the blockwise Hessian spectra and the full Hessian spectrum.

- **GPT2 (125M) on Openwebtext [11].** We use the code base of NanoGPT. We use batch size $= 245,760$ tokens. Due to the large number of parameters, we are not able to calculate the blockwise Hessian spectra for all parameter blocks. Instead, we apply SLQ to: the embedding layer; the output layer; the 1-st, 4-th, 8-th, 12-th attention blocks; and the 1-st, 4-th, 8-th, 12-th MLP blocks (note that the 12-th attention and MLP blocks are the final ones). In total, it takes

---

[7] https://github.com/pytorch/examples/blob/main/imagenet/main.py

[8] https://github.com/huggingface/pytorch-image-models

[9] https://medium.com/data-and-beyond/complete-guide-to-building-bert-model-from-sratch-3e6562228891

[10] https://github.com/karpathy/nanoGPT/

[11] https://huggingface.co/datasets/Skylion007/openwebtext

about 7 days on one A100 GPU to estimate all the blockwise Hessian spectra and the full Hessian spectrum.

**Training configuration.** In all cases, we train all the models under the default configurations in the above codebase. We grid-search the learning rates for SGD and Adam under the same budget and report the best result for each optimizer. We use the cosine-decay learning rate schedule for vision tasks. For SFT task, we use nanoGPT codebase. We first pre-train GPT2 on OpenwebText for 25B tokens and then fine-tune it on a subset of Alpaca Eval [12].

## D.2 Ablation Studies of SLQ on a Small Tranformer

On a small GPT model with 75k parameters, we compare (1) the true Hessian spectrum and (2) the estimated Hessian spectrum curve by SLQ method. As shown in Figure 16, we find that the SLQ method can produce accurate estimation.

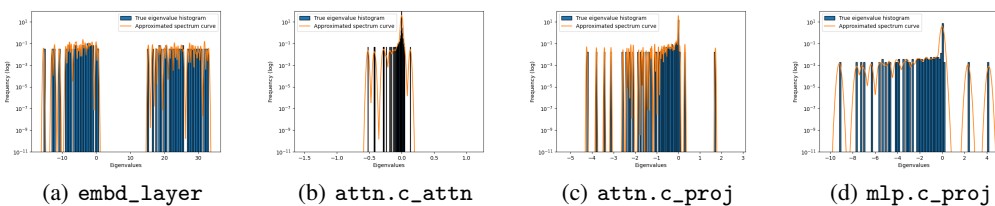

(a) `embd_layer`    (b) `attn.c_attn`    (c) `attn.c_proj`    (d) `mlp.c_proj`

Figure 16: (a, b, c, d): The comparison of (1) the true Hessian spectrum; and (2) the estimated Hessian spectrum curve by SLQ method. The experiments are conducted on a small GPT model with 75k parameters. We find that the SLQ method can produce accurate estimation.

## D.3 Implementation Details on Figure 2

We employ a synthetic dataset designed for binary classification, with 100 data points generated through the process outlined below. Our model is a 1-hidden-layer neural network, featuring an input size of 64 and a layer width of 8, utilizing the hyperbolic tangent (`Tanh`) as the activation function. We train this model over 1000 iterations using the Adam optimizer with a learning rate set to $1 \times 10^{-4}$, achieving the classification accuracy of 100%.

```python
def generate_data(n_samples_per_class, n_classes, input_dim):
    # Generate synthetic data for specified dimensions
    X = []
    y = []
    for i in range(n_classes):
        center = np.random.rand(input_dim) * 10  # Random class center
        class_samples = np.random.randn(n_samples_per_class, input_dim
    ) * 0.5 + center  # Add some noise
        X.append(class_samples)
        y.extend([i] * n_samples_per_class)

    X = np.vstack(X)   # Combine all class samples
    y = np.array(y)     # Convert labels to a NumPy array
    return X, y
```

## D.4 Implementation Details on the MLP experiments in Figure 5

We train a 4-layer MLP on MNIST. We use batch size = 128 and width = 300, 128, and 64 for the hidden layers. We use ReLU activation. We change the degree of heterogeneity by scaling the output of each layer with constant $c \in \mathbb{N}$. We scale $c$ from 1 to 15. For each $c$, we train SGD and Adam with default hyperparameters by grid-searching the learning rate from 1e-4 to 1e-1 and report the best test accuracy after 1 epoch.

---

[12]https://huggingface.co/datasets/tatsu-lab/alpaca_eval

# E Proofs

## E.1 Proof of Proof of Proposition 1

Let $H = \begin{bmatrix} L & 0 \\ 0 & \mu \end{bmatrix}$, where $L > \mu > 0$. We choose the initial point as $w^0 = (w_1^0, w_2^0) = (\sqrt{\mu/L}, \sqrt{L/\mu})$. By the update rule of GD, we have

$$
\begin{aligned}
\mathcal{L}(w^{t+1}) &= \mathcal{L}\left(w^t - \eta \nabla \mathcal{L}(w^t)\right) \\
&= \frac{1}{2}(w^t - \eta H w^t)^T H (w^t - \eta H w^t) \\
&= (w_1^t)^2 |1 - \eta L| L + (w_2^t)^2 |1 - \eta \mu| \mu \\
&= |1 - \eta L|^t L \frac{\mu}{L} + |1 - \eta \mu|^t \mu \frac{L}{\mu} \\
&= \mu |1 - \eta L|^t + L |1 - \eta \mu|^t
\end{aligned}
\tag{11}
$$

To proceed, we discuss the following cases:

When $\eta \leq 1/L$, since $|1 - \eta L|^t$ and $|1 - \eta \mu|^t$ are monotonically decreasing , the optimal solution is $\eta = 1/L$.

When $\eta \geq 1/\mu$, since $|1 - \eta L|^t$ and $|1 - \eta \mu|^t$ are monotonically increasing , the optimal solution is $\eta = 1/\mu$.

When $1/L \leq \eta \leq 1/\mu$, (11) can be written as $g_t(\eta) = \mu(\eta L - 1)^t + L(1 - \eta\mu)^t$. Take the first-order and the second-order derivative of the g, we can obtain $g_t'(\eta) = tL\mu(\eta_L - 1)^{t-1} - t\mu L(1 - \eta\mu)^{t-1}$ and $g_t''(\eta) = t(t-1)L^2\mu(\eta L - 1)^{t-2} + t(t-1)\mu^2(1 - \eta\mu)$. Since $g_t''(\eta) \geq 0$ for all $\eta \in [1/L, 1\mu]$, the function g is convex. By solving the equation that $g_t'(\eta) = 0$, we can obtain $\eta = \frac{2}{L+\mu}$ is a solution for all $t$. Plugging this result into (11) and rearranging the terms, we conclude the proof of Proposition 1.

## E.2 Proof of Theorem 1

We first show that $C_{l,2}$ and $C_{l,1}$ are non-zero w.p.1. under the random initialization in Assumption 1. We define set $S_i = \{w; h_i^T w = 0\}$ where $h_i \in \mathbb{R}^d$ is the $i$-th row of $H$. Since $H$ is positive definite, there is at least one non-zero entry in $h_i$, $i \in [d]$. As such, $S_i$ is a $(d-1)$-dimensional subspace of $\mathbb{R}^d$ and thus has zero Lebesgue measure in $\mathbb{R}^d$. Since $w^0$ follows continuous distribution, we have $\Pr\left(\{w^0; h_i^T w^0 = 0\}\right) = 0$, for $i = [d]$. Then we have

$$
\begin{aligned}
\Pr\left(\nabla \mathcal{L}(w^0) \text{ has at least one zero entry}\right) &= \Pr\left(H w^0 \text{ has at least one zero entry}\right) \tag{12} \\
&= \Pr\left(\cup_{i=1}^d \{w^0; h_i^T w^0 = 0\}\right) \tag{13} \\
&\leq \sum_{i=1}^d \Pr\left(\{w^0; h_i^T w^0 = 0\}\right) \tag{14} \\
&= 0. \tag{15}
\end{aligned}
$$

Therefore, $\nabla \mathcal{L}(w^0)$ is elementwise non-zero w.p.1.., so $C_{l,1}$ and $C_{l,2}$ are non-zero for all $l \in [L]$, w.p.1.. In the following analysis, We will assume $C_{l,1}$ and $C_{l,2}$ are non-zero.

Without loss of generality, we assume $h = 0$. This is because minimizing $\mathcal{L}(w) = \frac{1}{2} w^T H w - h^T w$ is equivalent to minimizing $\mathcal{L}(w) = \frac{1}{2}\left(w - w^*\right)^T H \left(w - w^*\right)$ where $w^* = H^{-1} h$. By a linear transformation $z = w - w^*$, Adam for minimizing $\frac{1}{2}\left(w - w^*\right)^T H \left(w - w^*\right)$ starting from $w^0$ is equivalent to Adam for minimizing $\frac{1}{2} z^T H z$ starting from $z^0 = w^0 - w^*$. Thus we can assume $w^* = 0$, or equivalently, $h = 0$. The update rule of Adam becomes

$$w^{t+1} = w^t - \eta (D^0_{Adam})^{-1} H w^t,$$

where $D^0_{Adam} = \text{diag}(\nabla \mathcal{L}(w^0) \circ \nabla \mathcal{L}(w^0))^{\frac{1}{2}} = \text{diag}(|H w^0|)$. We denote $d_t = \eta (D^0_{Adam})^{-1} H w^t$ and thus we have $w^t = \frac{1}{\eta} H^{-1} D^0_{Adam} d^t$ and $w^{t+1} = w^t - d_t$. These relations also hold for each block by changing the notation to $H_l w_l^t$, $D^0_{Adam}$, and $d_l^t$, etc.. Following the framework in [80], we try to bound the error yet to be optimized (a.k.a., cost-to-go) and the per-step improvement, respectively. The ratio of these two terms characterizes the rate of convergence. We now express both terms using $d_l^t$. For the cost-to-go term for the $l$-th block, we have

$$[\mathcal{L}(w^t)]_l - [\mathcal{L}^*]_l = \frac{1}{2}(w_l^t)^T H_l w_l^t = \frac{1}{2\eta^2}(d_l^t)^T D^0_{Adam,l} H_l^{-1} D^0_{Adam,l} d_l^t. \tag{16}$$

For the per-step improvement, we have

$$
\begin{aligned}
[\mathcal{L}(w^t)]_l - [\mathcal{L}(w^{t+1})]_l &= \frac{1}{2}(w_l^t)^T H_l w_l^t - \frac{1}{2}(w_l^{t+1})^T H_l w_l^{t+1} \\
&= \frac{1}{2}(w_l^t)^T H_l w_l^{t+1} - \frac{1}{2}(w_l^t - d^t)^T H_l (w_l^t - d_l^t) \\
&= (d_l^t)^T H_l w_l^t - \frac{1}{2}(d_l^t)^T H_l d_l^t \\
&= \frac{1}{2}(d_l^t)^T \left( \frac{2}{\eta} D^0_{Adam,l} - H_l \right) d_l^t. \tag{17}
\end{aligned}
$$

To proceed, we denote $\hat{H} = (D^0_{Adam})^{-1} H$ and we denote its eigenvalues as $\hat{\lambda}_1 \geq \hat{\lambda}_2 \geq \cdots \hat{\lambda}_d$. Similarly, we denote $\hat{H}_l = (D^0_{Adam,l})^{-1} H_l$ and its eigenvalues $\hat{\lambda}_{l,1} \geq \hat{\lambda}_{l,2} \geq \cdots \hat{\lambda}_{l,d_l}$. Let $\eta = \min_{l \in [L]} C_{l,1}$, we have

$$
\begin{aligned}
\frac{[\mathcal{L}(w^t)]_l - [\mathcal{L}^*]_l}{[\mathcal{L}(w^t)]_l - [\mathcal{L}(w^{t+1})]_l} &= \frac{\frac{1}{\eta^2}(d_l^t)^T D^0_{Adam,l} H_l^{-1} D^0_{Adam,l} d_l^t}{(d_l^t)^T \left( \frac{2}{\eta} D^0_{Adam,l} - H_l \right) d_l^t} \\
&\leq \left\| \frac{1}{\eta^2} \left( \frac{2}{\eta} D^0_{Adam,l} - H_l \right)^{-1} D^0_{Adam,l} H_l^{-1} D^0_{Adam,l} \right\|_2 \tag{18} \\
&\overset{(*)}{\leq} \frac{C_{l,2}^2 \lambda_{l,1}^2}{(\min_{l \in [L]} C_{l,1}^2) \lambda_{l,1} \lambda_{l,d_l}} \tag{19} \\
&\leq \frac{\max_{l \in [L]} C_{l,2}^2}{\min_{l \in [L]} C_{l,1}^2} \kappa_l, \tag{20}
\end{aligned}
$$

where $(*)$ is due to: by Assumption 1, $D^0_{Adam,l} \preccurlyeq C_{l,2} \lambda_{l,1} I$, $\frac{2}{\eta} D^0_{Adam,l} - H_l \succcurlyeq \left( \frac{2}{C_{l,1}} C_{1l} \lambda_{l,1} - \lambda_{l,1} \right) I \succcurlyeq \lambda_{l,1} I$, where $\preccurlyeq$ and $\succcurlyeq$ are matrix inequalities. By rearranging both sides of (20), we have $[\mathcal{L}(w^{t+1})]_l - [\mathcal{L}^*]_l \leq \left( 1 - \frac{1}{\left( \frac{\max_{l \in [L]} C_{l,2}^2}{\min_{l \in [L]} C_{l,1}^2} \right) \kappa_l} \right) ([\mathcal{L}(w^t)]_l - [\mathcal{L}^*]_l)$. Summing up both sides over $l \in [L]$ and we conclude the proof.

$$
\begin{aligned}
\mathcal{L}(w^{t+1}) - \mathcal{L}^* &= \sum_{l=1}^L \left( [\mathcal{L}(w^{t+1})]_l - [\mathcal{L}^*]_l \right) \\
&\leq \sum_{l=1}^L \left( 1 - \frac{1}{\left( \frac{\max_{l \in [L]} C_{l,2}^2}{\min_{l \in [L]} C_{l,1}^2} \right) \kappa_l} \right) \left( [\mathcal{L}(w^t)]_l - [\mathcal{L}^*]_l \right)
\end{aligned}
$$

$$\leq \max_{l \in [L]} \left( 1 - \frac{1}{\left( \frac{\max_{l \in [L]} C_{l,2}^2}{\min_{l \in [L]} C_{l,1}^2} \right) \kappa_l} \right) \sum_{l=1}^{L} \left( [\mathcal{L}(w^t)]_l - [\mathcal{L}^*]_l \right)$$

$$= \max_{l \in [L]} \left( 1 - \frac{1}{\left( \frac{\max_{l \in [L]} C_{l,2}^2}{\min_{l \in [L]} C_{l,1}^2} \right) \kappa_l} \right) \left( \mathcal{L}(w^t) - \mathcal{L}^* \right).$$

