# OpenReview forum: "Why Transformers Need Adam: A Hessian Perspective"
_NeurIPS.cc/2024/Conference — NeurIPS 2024 poster_

### Official Review · Reviewer_A8Vc · 2024-07-11

**Soundness:** 3
**Presentation:** 4
**Contribution:** 3
**Rating:** 6
**Confidence:** 3

**Summary:**

This paper explores the performance gap between SGD and Adam when training transformers. The authors analyze the eigenspectrum of the Hessian in different neural network architectures and attribute the failure of SGD to the different Hessian spectra in transformers compared to CNNs and MLPs. They empirically show that while the full Hessian spectrum is similar at initialization across architectures, transformers exhibit block heterogeneity, meaning different parameter blocks (e.g., key/query layer versus MLP layer) have distinct spectra. Conversely, other architectures such as CNNs and MLPs are block homogeneous. By artificially inducing block heterogeneity in MLPs, they demonstrate through numerical experiments that SGD's performance declines as the block spectrum variation increases, while Adam's performance remains stable. They attribute Adam's success in block heterogeneous cases to its ability to assign different learning rates to different parameters. The authors also propose using the Jensen-Shannon distance of Hessian blocks as a quantitive measure of the performance gap. They also provide a preliminary theoretical analysis comparing the convergence rates of SGD and Adam on a quadratic function with a block-form Hessian, highlighting Adam's dependence on the fine-grained condition number of individual Hessian blocks.

**Strengths:**

This paper tackles a key question: understanding the optimization challenges of transformers with SGD and Adam is crucial for developing more efficient techniques for training large language models.

The paper is well-organized and easy to read. I find the ideas introduced throughout this work interesting and potentially useful to the community.

**Weaknesses:**

Quantifying the performance gap between SGD and Adam using the average JS distance between block-wise Hessian spectra is an interesting idea. However, based on the runtime reported in the appendix, the Hessian approximation itself seems computationally expensive, which can raise concerns about how practical it is for large models, especially since the experiments in this paper are only on small models.

Additionally, the main theoretical analysis is conducted in a pretty simplified setup where the momentum part of Adam is fully ignored ($\beta_1=0$ in Algorithm 3).

**Questions:**

1. Can you comment on the computational complexity of calculating the JS distance and the Hessian approximation? Specifically, can you compare the runtime of computing this quantitative measure to the training time of the models in your paper? For significantly larger models, do you expect this approach to scale efficiently and be more practical than training the model?

2. Is the architecture the only factor affecting the Hessian structure? I am curious if the training distribution, particularly label imbalance, also impacts the Hessian structure, since in some previous works previous works, such as [46], the performance gap has been attributed to the heavy-tailed distribution of labels in language data.

**Limitations:**

Yes

---

> ### Author Rebuttal · Authors · 2024-08-05
>
> Thanks for the insightful question and careful evaluation of our paper. We provide response as follows.
>
> **1. In theoretical analysis, the momentum part of Adam is fully ignored  ( $\beta_1=0$ in Algorithm 3).**
>
> Thanks for the comment. The current analysis ignores momentum because we want to focus on the benefit of assigning coordinate-wise lr, so we remove the effect of all other components to control variables. Nevertheless,  we agree that choosing $\beta_1=0$ simplifies the problem. However, exploring the role of momentum is an independent and equally challenging problem. We consider it as an important future direction.
>
>
>
> **2. On the computational complexity of calculating the JS distance and the Hessian approximation; compare runtime to the training time; scale efficiently to larger models**;
>
> Thanks for the great question. Currently, our JS distance is indeed rather expensive to compute: it requires comparable to one training run.  Fortunately,  we find the current deployment of SQL is redundant for measuring hessian heterogeneity. We propose the following simple tweaks to significantly reduce the computation time, while still effectively detecting the Hessian heterogeneity. We call it **simplified SQL**:
>
> 1. Change the hyperparemters of SQL, including:
> - 1-1: we change num_v = 10 to num_v = 1. In SQL, num_v decides the number of random Gaussian vectors to approximate the expected quadrature. It is reasonable to reduce num_v because in high dimensional space, random vectors tend to concentrate around their mean, so one random sample can already be informative enough.
> - 1-2: we change the Lanzcos step m = 100 to m = 10. The reduction on Lanzcos step will have a coarse estimation on the middle eigenvalue, but won't affect much the heterogeneity, which is more dependent on the extreme eigenvalues
> 2. Randomly sample a subset of blocks and reduce batch size for estimating the spectrum. We uniformly sample 50% blocks and choose batch size = 32 (previously batch size = 1024).
>
> We report the result and runtime in the Table below.  As a result, the simplified SQL can obtain the same message as the original SQL: **JS0 of ResNet is about 100x smaller than that of BERT.**  Further, the simplified SQL is highly efficient to compute. With this simplified SQL, we believe our method can efficiently scale to larger models.
>
> |  Model   |        JS0         | Time for JS0 | Time for Training | Time for JS0 / Time for training |
> | :------: | :----------------: | :----------: | :---------------: | :------------------------------: |
> |   BERT   | 98.83440094176784  |     20s      |        4h         |           0.001388889            |
> | ResNet18 | 0.3568875006145019 |     65s      |       87.5h       |           0.000206349            |
>
> Tested on: single V100
>
> **3. Is the architecture the only factor affecting the Hessian structure? I am curious if the training distribution, particularly label imbalance, also impacts the Hessian structure**
>
> We agree that there exist other factors (such as data distribution and label imbalance) that affect the Hessian structure.  However,  for ViT on ImageNet,  we find SGD (even after careful tuning) is still largely worse than Adam. As such, we believe architecture plays a more crucial rule.

---

> > ### Comment · Reviewer_A8Vc · 2024-08-11
> >
> > Thanks for your response. I think the results will interest the community. I'll keep my score as is.

---

### Official Review · Reviewer_tJ3K · 2024-07-12

**Soundness:** 2
**Presentation:** 3
**Contribution:** 2
**Rating:** 6
**Confidence:** 2

**Summary:**

This paper investigates why Adam outperforms SGD in Transformers by examining Hessian estimations. While the overall Hessian spectrum appears similar in both Transformers and other neural networks, the block-wise Hessian spectrum is heterogeneous across blocks in Transformers and homogeneous in other neural networks. The authors empirically confirm that this heterogeneity hampers SGD's performance through experiments involving Transformers, CNNs, MLPs, and quadratic problems. They attribute Adam's advantage to its coordinate-wise learning rate and provide initial theoretical analysis in a simplified setting.

**Strengths:**

- The connection between Adam's performance and Hessian block-wise heterogeneity is novel, as are the related empirical observations.
- Related work is comprehensively cited in the Appendix.
- Although the finding itself is not groundbreaking, it suggests potential for future research in theoretical analysis and practical improvements of Adam, particularly regarding its advantage in adaptively updating different blocks.

**Weaknesses:**

- The experiments support the claim that Adam outperforms SGD when Hessian block-wise heterogeneity structure is present. However, it is unclear whether this correlation implies causation beyond intuitive understanding. In general, the conceptual contribution is not very strong.
- The definition of "block" also seems arbitrarily determined. For example, Fig. 3 shows that all convolution layers have a similar spectrum, while the spectrum differs between Query, Value, and MLP blocks in Transformers. Given that one set is the same type of "block" (conv layer) and the other is not, this does not seem surprising. Since how the block is defined entirely determines Hessian heterogeneity versus homogeneity, this observation might be overfitting the transformer architecture and could be difficult to generalize to non-mainstream architectures as claimed.
- The case study on quadratics helps understand and attempt initial theoretical analysis, but it only partially represents behaviours in larger networks. The noise in SGD does not play a significant role here, and Adam with $\beta_2 = 0.99$ performs very differently than in larger networks.

**Questions:**

-  Including a more detailed experimental setting will help with reproducibility. The hyperparameters for experiments such as learning rate are not included. Since SGD is sensitive to hyperparameters, it's unclear whether the gap is due to the claimed reason or if hyperparams are insufficiently tuned. For example, the training curve of SGD in Figure 9(a) in ViT-ImageNet is much lower than Adam and that is not the case in [Figure 12, Kunstner et.al. Heavy-Tailed Class Imbalance and Why Adam Outperforms Gradient Descent on Language Models]. Can the author comment on this?
- Figure 1 shows the full spectrum across different time points to illustrate its evolution during training, yet all other plots in the paper related to the block-wise spectrum are presented at initialization. Is there a reason why the evolution of the block-wise spectrum is not shown?
- Are there any ablation studies on how well the Hessian spectrum is estimated through the SQL method, perhaps on reasonably small networks?
- Typo in Figure 7 y-axis label

**Limitations:**

Limitations are well-addressed

---

> ### Author Rebuttal · Authors · 2024-08-05
>
> Thanks for  the careful review of our work. Here is our respectful response.
>
> **1. It is unclear whether this correlation implies causation beyond intuitive understanding.**
>
> We provide the following evidence on causation.
>
> **I: Theory:** On quadratic models, our theory suggests that: when heterogeneity happens, the complexity of Adam is better than GD.
>
> **II: Control-variable experiments:**. We design the experiment as follows:
> 1) We take two Hessian matrices with an identical set of eigenvalues (cases 3 and 4 in Section 3.1). This ensures the performance of GD is fixed.
>
> 2) We re-arrange these eigenvalues in two different ways,  which lead to heterogeneity and homogeneity (case 3 and 4).
>
> 3)  We find Adam is faster when heterogeneity exists,  resulting in a huge gap with GD
>
> This experiment implies "heterogeneity" can cause the "gap between Adam and SGD."
>
>
> **2. Block seems arbitrarily determined.**
>
> We kindly point out that block is not arbitrarily determined. Here is our principle: we define "blocks" as the irreducible dense principal sub-blocks in Hessian.
>
> Since it requires heavy calculation on Hessian to precisely find blocks by the principle above,  we find a computationally-friendly way: We use the PyTorch partition to determine the blocks.  We find that partition in PyTorch can usually match our principle. For instance, in Figure 2, each block corresponds to the parameters in Q, K, V; and MLP, respectively.  Under this common partition strategy, we observe heterogeneity in Transformers but not in CNNs
>
> **3. This observation might be overfitting the transformer architecture.**
>
> We kindly point out that the observation is not overfitting to Transformers since we also observed heterogeneity on non-Transformer models such as MLP mixer (a real-world non-attention-based architecture, **Figure 5 in paper**), and SGD also performs worse than Adam here.
>
>
> **4. The case study on quadratics helps understand and attempt initial theoretical analysis, but it only partially represents behaviors in larger networks.**
>
> We agree that quadratic models cannot capture all behaviors of real NNs. To cover more behaviors in larger NN,  we need analysis on more fine-grained settings such as noisy quadratic models or 1-layer attention models.  It is an intriguing direction to pursue in the future.
>
> **5. It's unclear whether the gap is due to the claimed reason or if hyperparams are insufficiently tuned.**
>
> We believe the gap is due to Hessian heterogeneity, instead of improper hyperparam for SGD. For all Transformers,  we grid search the learning rate of SGD and report the best performance.  We grid-search lr of SGD as follows.
>
> - For BERT: we use lr  = 1e-4 for Adam.  For SGD, we search lr  over [1e-4, 1e-3, 1e-2, 1e-1]
>
> - For ViT: we use  lr  = 1e-4 for Adam.  For SGD, we search lr  over [1e-4, 5e-4, 1e-3, 5e-3, 1e-2, 5e-2, 1e-1]
>
> - For GPT2-nano: we use lr = 1e-2 for Adam, For SGD, we search lr  over [1e-5, 5e-5, 1e-4, 5e-4, 1e-3, 5e-3, 1e-2, 5e-2, 1e-1, 5e-1, 1]
>
> We visualize the performance of SGD under different lr.  This can be seen in **Figure 1 in the attached PDF**. We find that SGD (even after careful tuning) is still worse than Adam, including ViT on ImageNet.
>
> **Additional evidence for ViT:**  We here provide more evidence that "SGD is worse than Adam on ViT" from the literature.
>
> - On Cifar-10, [1] carefully tuned SGD on ViT and find that SGD cannot work (Figure 4). They reported that "the SGD algorithm did not perform well for this task” even after careful tuning: “ The optimizer hyperparam were set according to the values obtained during hyperparam tuning. ”
>
> - On ImageNet, [2] provides empirical evidence that SGD is worse than Adam on vanilla ViT. The authors reported that " SGD yields significantly worse results than AdamW (on ViT)", "ViT often fails to converge with SGD " (their Figure 3).
>
> As for Figure 12 in Kunstner et.al., there are two possible reasons for the mismatch: (1) they used SimpleViT, which is a simplified version of the vanilla ViT as we conducted. It is possible that SimpleViT exhibits less heterogeneity and is more friendly to SGD. (2) perhaps they did not tune Adam carefully enough and Adam can perform better if well-tuned.
>
> [1] AdaGC: A Novel Adaptive Optimization Algorithm with Gradient Bias Correction
>
> [2] Early convolutions help transformers see better
>
> **6. Block-wise spectrum along training.**
>
> Following your suggestions, we plot the block-wise spectrum at 25%, 50%, and 100% training steps for GPT2 and ViT.  We find an interesting phenomenon: Hessian heterogeneity tends to reduce along training. This can be seen in **Figure 2 and 3 in the attached PDF.**
>
> We also take the checkpoint of ViT at the 46th epoch (which is about 50% steps) and continue to train with SGD. We find SGD now can perform much better. We provide a similar explanation here:  SGD now performs better because there is less Hessian heterogeneity in ViT at the 46-th epoch (50\% training steps). This can be seen in **Figure 2 in the attached PDF.**
>
> This phenomenon above also provides one interesting practical implication: if we can find a good initialization with less heterogeneity, it is possible to train transformers with SGD. However, designing such initialization is non-trivial, as its explicit relation with heterogeneity is still unclear. We leave this topic as an intriguing future direction.
>
> To sum up, we make two new findings when investigating the blockwise spectrum along training:  1. Heterogeneity tends to reduce along training. 2. as heterogeneity is reduced, we can switch Adam to SGD. We will include these results in the revised script.
>
>
> **7. Ablation studies on how well the Hessian spectrum is estimated through the SQL method, perhaps on reasonably small networks?**
>
> We provide an ablation study on a 4-layer Transformer. The results are shown in **Figure 4 in the attached PDF.**  We find SQL provide accurate approximation over the true eigenvalue density.

---

> ### Comment · Reviewer_tJ3K · 2024-08-14
> **Response to authors**
>
> Thank you for addressing my concerns; I will raise my score accordingly.

---

### Official Review · Reviewer_pTmy · 2024-07-12

**Soundness:** 3
**Presentation:** 3
**Contribution:** 4
**Rating:** 7
**Confidence:** 3

**Summary:**

The paper investigates why SGD performs worse than Adam in Transformers from the lens of Hessian. They first find that Transformers are "heterogeneous", that is, the Hessian spectrum across parameter blocks varies dramatically. Then they conduct various tasks on Transformers, CNNs, MLPs, and quadratic problems, to verify that block heterogeneity hampers SGD. Finally, they derive some initial theoretical analysis to indicate that SGD fails because it applies one single learning rate for all blocks, which cannot handle the heterogeneity among blocks.

**Strengths:**

1. The paper is well written, clear to read, and the story is interesting.

2. The discovery of block heterogeneous for the reason of the bad performance of SGD on Transformers is interesting and novel.

3. The experiments are completed to verify the points described in the paper.

**Weaknesses:**

1. The paper demonstrates that for different models when there the block heterogeneous at initialization increases, SGD becomes much worse than Adam. However, these comparisons are between different models. It is natural to ask for the same Transformers model, when the initialization changes to induce different block heterogenous levels, how would the performance of SGD compared with Adam change?

2. The theoretical results of the convergence of Adam for the quadratic models show that homogeneous spectra have worse complexity than heterogeneous spectra, is this the truth in practice?

**Questions:**

See weakness

**Limitations:**

The authors have addressed the limitations.

---

> ### Author Rebuttal · Authors · 2024-08-05
>
> We are grateful for the careful review of our paper and the great questions. Please find our respectful reply below.
>
> **1. For the same Transformers model, when the initialization changes to induce different block heterogenous levels, how would the performance of SGD compared with Adam change?**
>
> Thanks for the interesting question. Here is our finding: For the same model, when heterogeneity is reduced, the gap between Adam and SGD is also reduced. We provide evidence as below.
>
> There are typically two approaches to change initialization (please let us know if neither of these aligns with your thoughts).
>
> - Approach 1: Change the variance of random initialization.
>
> - Approach 2: Choose a checkpoint in the middle of training as the initialization.
>
> Unfortunately, we find Approach 1 is not suitable for controlling the heterogeneity: it seems unclear how the variance of random initialization is explicitly related to the heterogeneity in Hessian. As such, we use Approach 2 to change the initialization and track the performance of SGD.  We present two experiments below.  (**Exp 1** is already presented in the paper; **Exp 2** is new.)
>
> **Exp 1:**  On finetuning tasks of GPT2, when we change the initialization to the pre-trained weights, GPT2 (pre-trained) exhibits much less hetereogeneity compared with GPT2 with random initialization. This can be seen in **Figure 4 (f) and Figure 6 (a) in the paper**.  As a result, SGD can reach a similar loss as Adam here. This is shown in **Figure 6 (b)** in the paper.
>
> **Exp 2:** We realize that **Exp 1** might not fully answer your question: although **Exp 1** does not change the architecture,  it changes the training dataset (from pre-training dataset to SFT dataset), and thus changes the loss function. To rigorously address your concern, we further take the checkpoint of ViT at the 46-th epoch (which is about 50% training steps) and calculate Hessian heterogeneity. We make two findings: (1) For ViT checkpoint at the 46-th epoch, Hessian heterogeneity is largely reduced compared with that of random initialization. (2 ) If we continue to rain it with SGD, we find SGD now can perform much better than training from scratch. This result can be seen in **Figure 2 in the attached PDF**.
>
> This phenomenon above also provides one interesting practical implication: if we can find a good initialization with less heterogeneity, it is possible to train transformers with SGD. Yet, as discussed above, designing such initialization is non-trivial,  as its explicit relation with heterogeneity is still unclear.  We leave this topic as an intriguing future direction.
>
> **2. The theoretical results of the convergence of Adam for the quadratic models show that homogeneous spectra have worse complexity than heterogeneous spectra, is this the truth in practice?**
>
> Thanks for the great question. Let us clarify: our theory does **not** imply that Adam has worse complexity on homogeneous spectra. For simplicity of discussion, let's state your comment as a conjecture.
>
> **Conjecture 1:** Adam on quadratic problems with homogeneous spectra has worse complexity than Adam on heterogeneous spectra.
>
> If we understand your question correctly  (please tell us if not so), your question states that our theory implies Conjecture 1. We kindly point out that Conjucture 1 is **not correct**. We'll clarify below.
>
> (1) **What we proved and why it does NOT imply Conjecture 1**.
> **Our theoretical result:** Adam has complexity $O(\max_l \kappa_l)$. GD has complexity $\Omega(\kappa)$
>
> If our result implies Conjecture 1, one needs the following argument: when changing heterogeneity to homogeneity, $\max_l \kappa_l$ increases, and thus Adam is slower.
>
> However, "changing heterogeneity to homogeneity" does **not** necessarily mean "$\max_l \kappa_l$  increases". Actually, "$\max_l \kappa_l$" **can change in an arbitrary way** (can increase, decrease, or keep the same) when changing the heterogeneity. See detailed examples below.
>
> (2) **Detailed Examples on Comparing Homo and Hetero Cases**.
> We provide three examples below.
>
> **Notation:** We use Adam (homo) to denote the rate of Adam on homogeneous Hessian. Similarly for Adam (hetero)
>
> **Example 1:  Adam (homo) is same as Adam (hetero)**
>
> case 1-1: homo: eigenvalue {1,2}, {1,2}
>
> case 1-2:  hetero: eigenvalue {1,2}, {11,12}
>
> Since $\max_l \kappa_l$ are the same for both case 1-1 and 1-2, Adam (homo) is same as Adam (hetero)
>
>
> **Example 2: Adam (homo) is faster than Adam (hetero)**
>
> case 2-1: homo: eigenvalue {1,1.5}, {1,1.5}
>
> case 2-2:  hetero: eigenvalue {1,2}, {11,12}
>
> Since case 2-1 has smaller  $\max_l \kappa_l$ than case 2-2, Adam (homo) is faster than Adam (hetero)
>
>
>
> **Example 3: Adam (homo) is slower than Adam (hetero)**
>
> case 3-1: homo: eigenvalue {1,11}, {2,12}
>
> case 3-2:  hetero: eigenvalue {1,2}, {11,12}
>
> Since case 3-1 has larger $\max_l \kappa_l$ than case 3-2, Adam (homo) is slower than Adam (hetero)
>
> To sum up, there is no conclusion on "whether Adam under homogeneity is faster or slower than Adam under heterogeneity ".  Either case can happen.
>
>
> (3) **possible source of confusion:** We thought (correct us if not so) the confusion may come from the numerical examples (Case 3 and 4 in Section 3.1) in the paper. If comparing two figures, Adam (homo) in Case 3 is slower than Adam (hetero) in Case 4. But as argued above, this is just one example, and it does NOT show Adam (homo) is ALWAYS slower than Adam (hetero).
>
>
> As a result, all the above three examples can happen, and "Changing hetero to homo" does not necessarily mean "Adam becomes slower".

---

> > ### Comment · Reviewer_pTmy · 2024-08-13
> >
> > Thanks for the authors' responses. The authors carefully address my questions. I thank the authors for pointing out that either case can happen for Adam under homogeneity and heterogeneity. The confusion is indeed caused by Case 3 and 4 in Section 3.1, where the diagonal elements are the same, but in different order. Maybe such reordering of the same elements has some correlation with the algorithm's complexity. Overall, I think this work is interesting and has potential for the community, thus I would like to update my score to 7.

---

### Official Review · Reviewer_5ncB · 2024-07-26

**Soundness:** 3
**Presentation:** 3
**Contribution:** 2
**Rating:** 5
**Confidence:** 3

**Summary:**

In this work the authors assert that the Adam optimizer works well on Transformers while Stochastic Gradient Descent (SGD) does not, and they attempt to explain this phenomenon by inspecting the spectrum of Transformers (i.e., the eigenvalues of the model's Hessian matrix) and other models, such as convolutional networks, where SGD is competitive with Adam.  They show that the spectrum of Transformers and convolutional networks are similar, providing little insight.  The main innovation of the authors is to investigate the spectrum of individual components/layers of these networks - the so-called block spectrum - and the main contribution of the work is to show that the block spectrum is empirically correlated with the ineffectiveness of SGD relative to Adam.

**Strengths:**

1) The paper is generally clear and well-written - this is appreciated
2) The idea of inspecting the block-wise spectrum is potentially innovative
3) The empirical results presented by the authors are fairly convincing.

**Weaknesses:**

1) The premise of this paper is that SGD performs worse for attention-based models, and it is unclear to me if this is true.  I say this for two reasons:
a) The authors cite [45,56] on Line 16 after making this claim, presumably to support the claim.  However, these are quite old publications and neither of these papers specifically make this claim.  Are there other publications that make these claims?
b) The authors also provide their own empirical results (e.g., Fig. 3) to support this claim, however, these results seem highly insufficient to me to support this claim.  How did the authors choose/optimize the learning rate for SGD, and/or the learning rate schedule?

If SGD does not generally perform worse than attention models on Transformers, then it seems to me that the work has little scientific value, since the differences in the block spectrum are just inconsequential differences, or at least that their implications are different than suggested by the authors.  e.g., perhaps the authors could argue that the heterogeneous block spectra implies that SGD needs a lot more tuning to work well, for example.    I'd ask the authors to please address this point, and I will consider raising my rating.

2) Novelty of using block spectrum (potentially).  To me this is the main value of this work.  I'm not as familiar with this topic, and it is unclear to me how non-obvious the use of a block spectrum would be for analysis, or how surprising the results of using this approach are: perhaps these findings would be somewhat obvious to researchers familiar with this topic?    I will defer to other reviewers for this assessment.

**Questions:**

See "Weaknesses" for my questions.

**Limitations:**

yes.

---

> ### Author Rebuttal · Authors · 2024-08-05
>
> Thanks for the careful evaluation of our paper. We respectfully provide our response as follows.
> **1-1. Are there other publications that make these claims (SGD worse than Adam on attention-based model)?**
>
> Thanks for raising this comment. Yes, it is widely reported that SGD largely underperforms Adam on attention-based models, **even after careful tuning**. Here are some references:
>
> 1. [1] provides empirical evidence that SGD is worse than Adam on BERT, but not on SGD. On Bert, the authors comment that "Although hyperparameters for SGD are finetuned, a large performance gap is still observed" (their Figure 1)
>
> 2. [2] provides empirical evidence that SGD is worse than Adam on ViT. The authors reported that " SGD yields significantly worse results than AdamW (on ViT)", "ViT often fails to converge with SGD " ( their Figure 3)
>
> 3. [3] carefully tuned SGD on ViT and find that SGD cannot work (their Figure 4). They reported that " the SGD algorithm did not perform well for this task”"  even after careful tuning: “ The optimizer hyperparameters were set according to the values obtained during hyperparameter tuning. ”
>
> 4. [4] carefully tuned SGD on modern GPT models and find that SGD cannot work (their Figure 1).
>
> 5. Besides, main-stream large language models use Adam, including GPT3 [5], Llama series [6], Gemini series [7], etc. This serves as additional evidence that SGD is worse than Adam for Transformer training.
>
>
> **1-2. How did the authors choose/optimize the learning rate for SGD, and/or the learning rate schedule?**
>
> For the learning rate schedule, we use the same default cosine decay with warmup for both SGD and Adam. As for the learning rate, we grid search the learning rate of SGD and report the best performance. We grid-search the learning rate as follows.
>
> - For BERT: we use lr  = 1e-4 for Adam.  For SGD, we grid search lr  over [1e-4, 1e-3, 1e-2, 1e-1]
>
> - For ViT: we use lr  = 1e-4 for Adam.  For SGD, we grid search lr  over [1e-4, 5e-4, 1e-3, 5e-3, 1e-2, 5e-2, 1e-1]
>
> - For GPT2-nano: we use lr = 1e-2 for Adam, For SGD, we grid search lr  over [1e-5, 5e-5, 1e-4, 5e-4, 1e-3, 5e-3, 1e-2, 5e-2, 1e-1, 5e-1, 1]
>
> We also visualize the performance of SGD under different learning rates.  This can be seen in **Figure 1 in the attached PDF**.  For all these Transformers we investigate, we find  SGD (even after fine-tuning carefully) performs significantly worse than Adam.
>
> **2. Novelty of using block spectrum (potentially).**
>
> Thanks for the question. We believe our idea is non-trivial and non-obvious. The novelty of our perspective is also confirmed by other three reviewers, for instance:
>
> - Reviewer PTmy: "The discovery of block heterogeneous for the reason of the bad performance of SGD on Transformers is interesting and novel."
>
> - Review tJ3K: "The connection between Adam's performance and Hessian block-wise heterogeneity is novel, as are the related empirical observations."
>
> - Reviewer A8Vc: " I find the ideas introduced throughout this work interesting and potentially useful to the community."
>
> **Our comment on why blockwise spectrum is non-trivial.** For completeness, we further comment on why the perspective of the blockwise spectrum is new and non-trivial.
>
> **--The natural idea does not work.** To explore "why Transformers need Adam", a natural idea is to investigate the full Hessian eigenvalue (spectrum). This is because: by optimization theory, full Hessian eigenvalues largely decide the behavior of gradient methods, and there is an active line of work trying to understand NN training via full Hessian spectrum (e.g., reference [8] here, also see the reference [12,32,39,67,68,69,75,76,97,98,103] in our paper ).  Unfortunately, we find there is no noticeable difference between the full spectrum of CNNs and that of Transformers (Figure 1 in the paper).  As such, the natural idea of full spectrum does not work.
>
> **--Why our perspective is non-trivial.**  Our major conceptual innovation is that we connect the following findings to "why Transformers need Adam".
>
> 1. Transformers contains various kinds of building blocks, while CNN consists of similar convolutional layers. Based on this, we conjecture that blockwise differences might be crucial.
>
> 2. "Hessian of NNs have near-block-diagonal structure" (a non-trivial but highly overlooked finding in this field). Based on this, we realize that the blockwise spectrum has rich information and could be used to quantify the blockwise difference. Note that for models with dense Hessian, blockwise spectrum can still be computed, but loses substantial information.
>
> As such, our blockwise spectrum perspective is novel to the deep learning community because it is based on multiple non-trivial findings.
>
> We believe this perspective is also new to optimization community: for optimization community,  it is very rare to analyze (near-) block-diagonal Hessian structure since typical problems do not have such structure.  For instance, in the classical non-linear programming dataset [9], all problems have non-block-diagonal Hessian.  We point out a new perspective to characterize modern optimization problems.
>
> As such, we believe our perspective is new, non-trivial, and potentially useful to a wide range of audiences.
>
> [1] Why are adaptive methods good for attention models?
>
> [2] Early convolutions help transformers see better.
>
> [3] AdaGC: A Novel Adaptive Optimization Algorithm with Gradient Bias Correction
>
> [4] Deconstructing What Makes a Good Optimizer for Language Models
>
> [5] Language models are few-shot learners
>
> [6] Llama 2: Open foundation and fine-tuned chat models
>
> [7] Gemini: A natural language system for spoken-language understanding
>
> [8] An investigation into neural net optimization via hessian eigenvalue density
>
> [9] Nonlinear Programming Solvers for Unconstrained and Constrained Optimization Problems: a Benchmark Analysis

---

### Author Rebuttal · Authors · 2024-08-05

Dear reviewers and AC:

We attached a PDF with the following four figures. Please check.

**Figure 1:** On ViT, BERT, and GPT2-nano, we carefully grid search the learning rate for SGD and report all the results. We find that on all these tasks, SGD (even after careful tuning) is significantly worse than Adam.

**Figure 2 and 3:**  For ViT and GPT2, we plot the evolution of heterogeneity of Hessian along training. We find that heterogeneity is reduced along with training. Further, when switching Adam to SGD in the middle of training where the heterogeneity is reduced (e.g., the 46th epoch of ViT training), SGD can perform better.

**Figure 4:**  We conduct ablation study on small GPT to show that our SQL method can accurately calculate the Hessian spectrum.

---

### Comment · Area_Chair_fiSa · 2024-08-13
**TLDR: Reviewers please do acknowledge the rebuttals and react to them.**

Dear reviewers,

thanks for your reviews.  Please do look at the rebuttals of the author and give the authors some feedback whether they could address your concerns.  This is really important for the authors.

Best regards Your AC.

---

### Decision · Program_Chairs · 2024-09-25

**Decision:**

Accept (poster)

**Comment:**

Great paper!  Review scores: 6 6 7 5

The paper explores and partially answers the question why SGD is worse than Adam when training transformer models.  The reviews did identify potential weaknesses which the rebuttal of the authors did successfully clear out.  I didn't find the weaknesses pointed out by review with the lowest score (the 5, reviewer 5ncB) very convincing.  Also the review got a detailed rebuttal  as well, but that wasn't acknowledged by the reviewer.

So overall this is a really nice paper that should be accepted!